# Synaptotagmin-7 places dense-core vesicles at the cell membrane to promote Munc13-2- and Ca$^{2+}$-dependent priming

**Bassam Tawfik[1], Joana S Martins[1†], Sébastien Houy[1†], Cordelia Imig[2†‡], Paulo S Pinheiro[1,3], Sonja M Wojcik[2], Nils Brose[2], Benjamin H Cooper[2], Jakob Balslev Sørensen[1]\***

[1]Department of Neuroscience, University of Copenhagen, Copenhagen, Denmark; [2]Department of Molecular Neurobiology, Max Planck Institute of Experimental Medicine, Göttingen, Germany; [3]Center for Neuroscience and Cell Biology, University of Coimbra, Coimbra, Portugal

**Abstract** Synaptotagmins confer calcium-dependence to the exocytosis of secretory vesicles, but how coexpressed synaptotagmins interact remains unclear. We find that synaptotagmin-1 and synaptotagmin-7 when present alone act as standalone fast and slow Ca$^{2+}$-sensors for vesicle fusion in mouse chromaffin cells. When present together, synaptotagmin-1 and synaptotagmin-7 are found in largely non-overlapping clusters on dense-core vesicles. Synaptotagmin-7 stimulates Ca$^{2+}$-dependent vesicle priming and inhibits depriming, and it promotes ubMunc13-2- and phorbolester-dependent priming, especially at low resting calcium concentrations. The priming effect of synaptotagmin-7 increases the number of vesicles fusing via synaptotagmin-1, while negatively affecting their fusion speed, indicating both synergistic and competitive interactions between synaptotagmins. Synaptotagmin-7 places vesicles in close membrane apposition (<6 nm); without it, vesicles accumulate out of reach of the fusion complex (20–40 nm). We suggest that a synaptotagmin-7-dependent movement toward the membrane is involved in Munc13-2/phorbolester/Ca$^{2+}$-dependent priming as a prelude to fast and slow exocytosis triggering.

**\*For correspondence:**
jakobbs@sund.ku.dk

[†]These authors contributed equally to this work

**Present address:** [‡]Department of Neuroscience, University of Copenhagen, Copenhagen, Denmark

## Introduction

Neurotransmitter or hormone release requires a basal membrane fusion machinery, and one or more Ca$^{2+}$-sensors to link fusion to the electrical activity of the cell. The machinery driving vesicle-to-membrane fusion consists of the SNAREs (*Fang and Lindau, 2014*; *Jahn and Fasshauer, 2012*), associated proteins Munc18 and Munc13 necessary for SNARE-complex assembly (*Rizo and Xu, 2015*), whereas the Ca$^{2+}$-sensors are proteins of the synaptotagmin (Syt) family (*Pinheiro et al., 2016*), which act together with complexins (*Makke et al., 2018*; *Trimbuch and Rosenmund, 2016*). Syts harbor two C2-domains, which can bind to Ca$^{2+}$ and phospholipids (in 8 of the 17 Syt isoforms present in the mammalian genome), and to SNAREs (*Südhof, 2002*). Analysis in expressing cells showed that Syt-1, Syt-2, and Syt-9 trigger fast, synchronous release (*Xu et al., 2007*). Synaptotagmin-7 (Syt-7) displays the slowest membrane binding/unbinding kinetics of all the synaptotagmins (*Hui et al., 2005*), and the highest Ca$^{2+}$-affinity during lipid binding (*Bhalla et al., 2005*; *MacDougall et al., 2018*; *Sugita et al., 2002*), although the apparent Ca$^{2+}$-affinities vary with the experimental system and lipid composition (in terms of half-maximal lipid-binding, Syt-7: 0.3–2 µM; Syt-1: 10–150 µM [*Bhalla et al., 2005*; *Sugita et al., 2002*]). Syt-7 is expressed in high amounts in the brain (*Sugita et al., 2001*) and the adrenal medulla (*Fukuda et al., 2004*; *Osborne et al., 2007*; *Schonn et al., 2008*) where it is typically found co-expressed with Syt-1. The co-existence of two Syts with very different kinetics and Ca$^{2+}$-affinities in the same cell raises the question whether they

act autonomously/additively, compete with each other, or cooperate in a shared mechanism (*Walter et al., 2011*).

When Syt-1 is deleted, fast release is eliminated and residual slow/asynchronous release is driven by Syt-7 (*Bacaj et al., 2013*; *Schonn et al., 2008*), which is consistent with an additive function of the two sensors. However, the function of Syt-7 in the presence of the faster Syt isoform (Syt-1 or Syt-2) is not clear. In the presence of Syt-1 or Syt-2, the removal of Syt-7 reduces asynchronous release following a single action potential in cerebellar granule cell synapses (*Turecek and Regehr, 2018*), but not in cultured glutamatergic neurons (*Liu et al., 2014*; *Weber et al., 2014*), in the Calyx of Held synapse (*Luo and Südhof, 2017*) or in cerebellar basket cell synapses (*Chen et al., 2017*). During high-frequency stimulation, Syt-7 adds a sustained or tonic component of release in a number of different depressing synapses (*Bacaj et al., 2013*; *Chen et al., 2017*; *Liu et al., 2014*; *Luo and Südhof, 2017*; *Turecek et al., 2017*; *Wen et al., 2010*). Reduced sustained and fast release was first described in the Syt-7 knock-out (KO) chromaffin cells (*Schonn et al., 2008*). Liu et al. suggested that Syt-7 acts as an upstream $Ca^{2+}$-sensor that speeds up vesicle recruitment during train stimulation, resulting in sustained release while the $Ca^{2+}$ concentration is high (*Liu et al., 2014*). The function in priming was supported by delayed calcium-dependent recovery after high frequency or sucrose stimulations in cultured glutamatergic hippocampal neurons from the Syt-7 KO (*Liu et al., 2014*). Conversely, Bacaj and collaborators did not find differences in recovery in Syt-7 KO hippocampal neurons; instead, they reported that Syt-7 acts together with Syt-1 to ensure full capacity of the primed vesicle pool, which implies a function in stabilizing primed vesicles (*Bacaj et al., 2015*). Syt-7 also supports asynchronous glutamate release from principal cells onto Martinotti cells (*Deng et al., 2020*) and Syt-7 is necessary for short-term synaptic facilitation (*Barthet et al., 2018*; *Jackman et al., 2016*). After prolonged stimulation trains, or in the presence of manipulations exacerbating asynchronous release, Syt-7 directs vesicles toward a slow endocytosis pathway (*Li et al., 2017*; *Virmani et al., 2003*). Thus, Syt-7 plays multiple roles in exocytosis and endocytosis (*Chen and Jonas, 2017*; *Huson and Regehr, 2020*; *MacDougall et al., 2018*).

Adrenal chromaffin cells were the first cells in which vesicle priming was shown to be calcium-dependent (*Bittner and Holz, 1992*; *von Rüden and Neher, 1993*), which was later found to apply also to neurons (*Dittman and Regehr, 1998*; *Gomis et al., 1999*; *Schneggenburger et al., 2002*; *Stevens and Wesseling, 1998*; *Wang and Kaczmarek, 1998*). Chromaffin cells also display both fast and slow release phases when stimulated by a strong stimulus, such as $Ca^{2+}$ uncaging (*Heinemann et al., 1993*). Here, we used adrenal chromaffin cells, mouse knockouts, electrophysiology, and high-resolution 3D electron tomography, to investigate whether in this cellular system the two $Ca^{2+}$-sensors Syt-1 and Syt-7 can be said to act independently, or whether they are interdependent – that is engaging in either cooperative or competitive interplay (*Walter et al., 2011*).

## Results

Chromaffin cells offer distinct advantages in the study of neurosecretion (*Neher, 2018*). Since chromaffin cells do not have a limited number of release sites, $Ca^{2+}$-dependent priming leads to a large and readily measurable increase in the size of the pool of primed large dense-core vesicles (LDCV) when prestimulation $[Ca^{2+}]$ exceeds 100–200 nM. When combined with intracellular $Ca^{2+}$-control via $Ca^{2+}$-uncaging, this makes it possible to accurately titrate the $Ca^{2+}$-dependence of priming in the steady state (*Voets, 2000*), something which has not been achieved in other cell types. Here, we define 'priming' as the reaction (re-)filling the releasable vesicle pools; a clear kinetic distinction between priming and fusion when triggered by a common stimulus ($Ca^{2+}$) requires that fusion is >5–10-fold faster than priming; this requirement is fulfilled using $Ca^{2+}$-uncaging, which increases the fusion rate abruptly. For these reasons, we used chromaffin cells and $Ca^{2+}$-uncaging to investigate the involvement of Syt-1 and Syt-7 in the $Ca^{2+}$-dependence of priming and fusion triggering.

### When present alone, Syt-1 and Syt-7 act as kinetically distinct fusion triggers

We first studied the function of each Syt isoform in isolation, by expressing Syt-1 and Syt-7 individually using lentiviral vectors (see Materials and methods) in chromaffin cells cultured from Syt-1/Syt-7 double KO (DKO) mouse embryos (*Schonn et al., 2008*). We stimulated secretion using $Ca^{2+}$-uncaging to abruptly raise the intracellular $Ca^{2+}$ concentration ($[Ca^{2+}]_i$) from the sub-μM range to around

20–30 µM (*Figure 1A*). This mode of stimulation causes LDCV fusion at a spatially homogeneous $[Ca^{2+}]_i$, and allows the distinction between fusion kinetics and vesicle pool size. Vesicle pool sizes are affected by priming reactions, which take place before arrival of the fusion trigger ($Ca^{2+}$), whereas fusion kinetics is determined by events downstream of $Ca^{2+}$ arrival. Exocytosis was monitored simultaneously by membrane capacitance measurements, which assesses the plasma membrane area, and amperometry, which measures oxidizable neurotransmitters (mainly adrenaline). Upon $Ca^{2+}$-uncaging, DKO cells displayed a very small response (*Figure 1A*; *Schonn et al., 2008*). In contrast, when

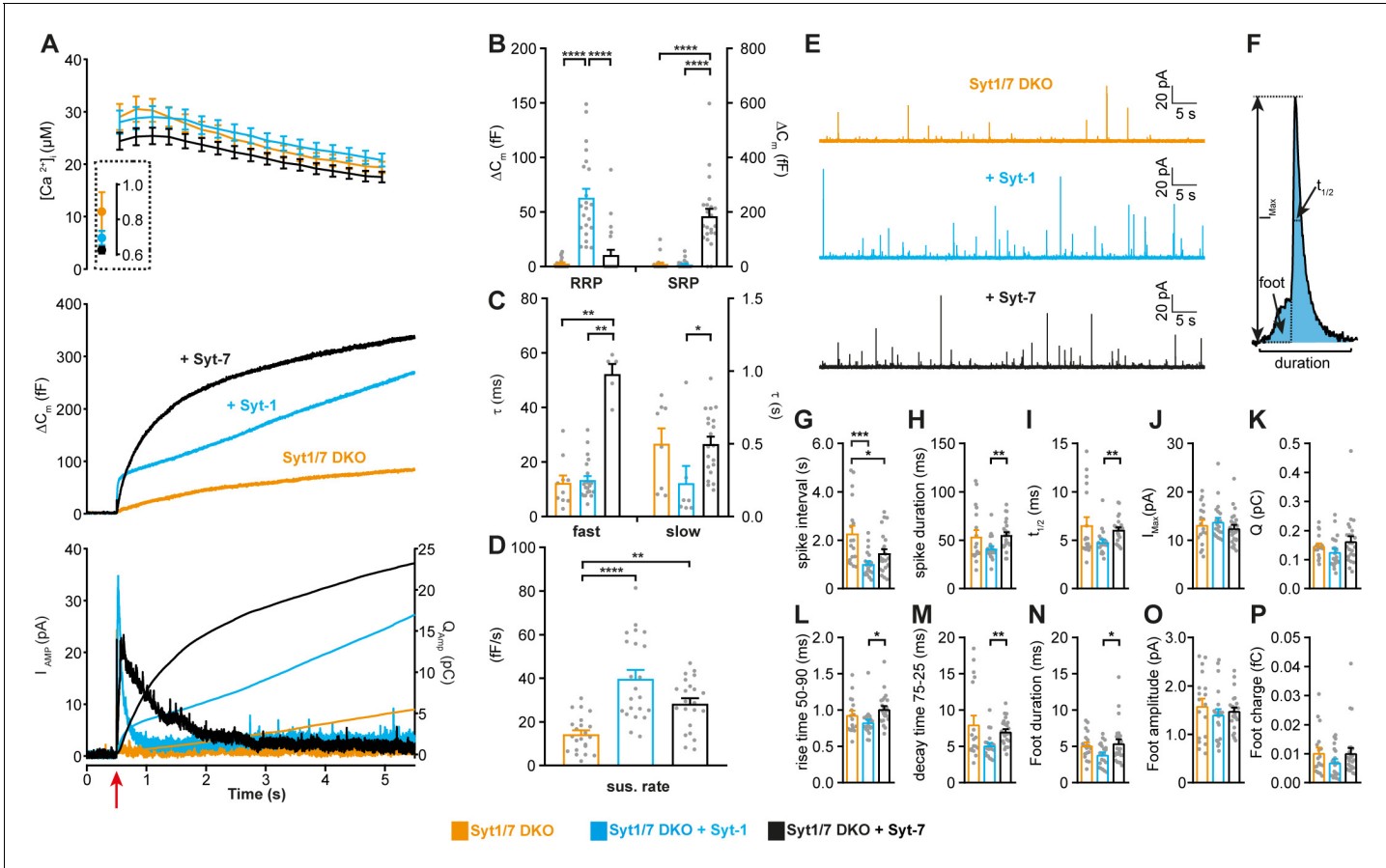

**Figure 1.** Syt-1 and Syt-7 are stand-alone calcium-sensors with different kinetics. (A) Calcium uncaging experiment in Syt-1/Syt-7 DKO cells (orange) and in DKO cells overexpressing Syt-7 (black) or Syt-1 (blue). Top panel: $[Ca^{2+}]$ before (insert) and after calcium uncaging (uncaging flash at red arrow, bottom panel). Middle panel: capacitance traces (mean of all cells) show that the secretion is potentiated more (higher amplitude) by Syt-7 expression, but the kinetics of the secretory burst is faster after Syt-1 expression. Bottom panel: Mean amperometry (left ordinate axis) and mean integrated amperometry (right ordinate axis). Note that the integrated amperometric traces agree very well with the capacitance traces. (B) Sizes of the RRP and SRP. (C) Time constants, τ, of fusion for fast (i.e. RRP) and slow (i.e. SRP) secretion. (D) Sustained rates of secretion. Data information: In (A–D), data with error bars are presented as mean ± SEM; in (A), the traces are the mean of all cells. *: p<0.05; **: p<0.01; ***: p<0.001; ****: p<0.0001. Kruskal-Wallis test with Dunn's post-hoc test. Number of cells, DKO: N = 23 cells; DKO + Syt-1: N = 22 cells; DKO + Syt-7: N = 21 cells. (E) Amperometric currents induced by infusion of ~5 µM $Ca^{2+}$ into the cell via a patch pipette. Syt-1/Syt-7 DKO cells, DKO cells expressing either Syt-1 (blue trace) or Syt-7 (black trace). (F) Single amperometric spike, indicating measurement of peak current ($I_{max}$), total charge (Q, by integration), duration at half maximum ($t_{1/2}$), and total duration of spike. The foot signal, which reports on the fusion pore before it expands, is indicated. (G) The spike interval. (H) Spike duration. (I) Duration at half maximum ($t_{1/2}$). (J) Peak current ($I_{max}$). (K) Total charge of foot and spike (Q). (L) Spike 50–90% rise time. (M) Spike 75–25% decay time. (N) Duration of foot signal. (O) Amplitude of foot signal. (P) Charge of foot signal. The spike interval was significantly decreased by expression of either Syt-1 or Syt-7 in DKO cells. The shape parameters show that spikes have faster dynamics in the presence of Syt-1 than with Syt-7. Data information: In (G–P), data are presented as mean ± SEM. *: p<0.05; **: p<0.01; ***p<0.001. In (G, L, N): One-way ANOVA with post-hoc Tukey's test. In (H, I, M): Kruskal-Wallis test with post-hoc Dunn's test. The spike interval (G) and the duration of foot signal (N) were log-transformed before statistical testing. Number of cells: DKO: N = 18 cells DKO + Syt-1: N = 21 cells; DKO + Syt-7: N = 24 cells.

The online version of this article includes the following figure supplement(s) for figure 1:

**Figure supplement 1.** Amperometric charge quantification.

either Syt-1 or Syt-7 was expressed, robust secretion resulted (*Figure 1A*). However, the kinetics of secretion were very different: Syt-1 supported a small rapid burst of secretion, with a fusion time constant around 10–20 ms (*Figure 1A–C*). In contrast, Syt-7 stimulated a larger burst of secretion, but with a much slower fusion time constant, around 500 ms (*Figure 1A–C*). These two components correspond kinetically to the previously identified Readily Releasable Pool (RRP) and Slowly Releasable Pool (SRP) (*Voets, 2000*). Fitting individual responses with a sum of two exponentials (representing the RRP and the SRP) and a linear function (for the sustained component) allowed us to identify the size and fusion kinetics of both the RRP and SRP (*Figure 1B–C*; see Materials and methods). Syt-1 supported fast secretion (i.e. an RRP, fusion time constant 13.4 ± 1.47 ms), but not slow-burst secretion (i.e. an SRP), whereas Syt-7 only supported slow secretion (SRP fusion time constant: 500 ± 48.9 ms; *Figure 1B–C*). The data also show that the slow burst became faster in the presence of Syt-1, and the fast burst was slower in the presence of Syt-7 (*Figure 1C*); thus, the effect of the Syts extend to both phases of release. The Syt-1/Syt-7 DKO apparently supported a very small RRP with fast time constant (*Figure 1B–C*); however, caution should be exercised when considering the kinetics and amplitude of a pool this small (3.4 ± 0.9 fF, corresponding to ~3.5 vesicles [*Pinheiro et al., 2014*]), because it can be affected by imperfections or noise in the capacitance measurements. Both isoforms supported a near-linear sustained release component (*Figure 1A,D*). *Figure 1A* (bottom panel) shows amperometric current and charge; note that the charge is consistent with the capacitance measurements, showing faster kinetics for Syt-1-driven release (quantification of total integrated amperometry is found in *Figure 1—figure supplement 1*). Thus, expression of either Syt-1 or Syt-7 can re-establish release in Syt-1/7 DKO cells and Syt-1 triggers faster fusion than Syt-7.

If the Syts act as stand-alone calcium-sensors, they might be expected to differently affect single-vesicle fusion events. This can be investigated by single-vesicle amperometry, where each fusion event is detected as a spike in the oxidation current (*Bruns, 2004*). Previous investigations in mouse chromaffin cells failed to find differences between Syt-1 KO and WT (*Voets et al., 2001*) or Syt-7 KO and WT spike shapes (*Segovia et al., 2010*). However, a difference between Syt-1 and Syt-7 might be detectable when comparing DKO cells overexpressing Syt-1 and Syt-7 side-by-side. To elicit sustained secretion at an intermediate frequency – so that single spikes can be resolved – we infused overexpressing DKO cells with an intermediate $Ca^{2+}$ concentration (~5 µM) via the patch pipette. A semi-automatic detection algorithm was used to identify spikes and quantify their shape (*Mosharov, 2008*). Expression of either Syt-1 or Syt-7 increased the spike frequency – or, equivalently, reduced the spike interval (*Figure 1E,G*). This increase in frequency was consistent with the increase in sustained released in uncaging experiments (*Figure 1D*); indeed, $Ca^{2+}$-infusion induces sustained release, but does not allow the build-up and abrupt fusion of standing pools of primed LDCVs. When analyzing the shape of the amperometric spikes, Syt-1 caused a shorter spike duration than Syt-7 (*Figure 1H*), a faster half-life (*Figure 1I*), a faster rise time (*Figure 1L*), a faster decay time (*Figure 1M*), and a shorter duration of the so-called 'foot' signal (*Figure 1N*). The foot signal measures the duration of the fusion pore prior to pore expansion (*Chow et al., 1992*). A longer foot signal following overexpression of Syt-7 compared to Syt-1 was previously found in PC12-cells (*Zhang et al., 2010*). These findings agree with and expand on data obtained using optical means (TIRF-microscopy), which showed a longer duration of the exocytotic event driven by Syt-7 (*Bendahmane et al., 2020*; *Rao et al., 2014*; *Rao et al., 2017*). Notably, with Syt-7 overexpression, spike kinetics was indistinguishable from the Syt-1/7 DKO condition. The $Ca^{2+}$-sensor fusing vesicles in the Syt-1/7 DKO is unknown, but a possible candidate is Doc2B (*Pinheiro et al., 2013*), which shares with Syt-7 a higher $Ca^{2+}$-affinity than that of Syt-1. Overall, these data establish the status of Syt-1 and Syt-7 as stand-alone fusion sensors and demonstrates that Syt-1 supports faster fusion than Syt-7 on both the population and the single-vesicle level.

## Syt-7 promotes $Ca^{2+}$-dependent priming of slow and fast release (SRP and RRP)

To study the impact of Syt-7 on release in the presence of Syt-1, we used Syt-7 KO chromaffin cells where endogenous levels of Syt-1 remained unchanged (see below: Figure 9B, Figure 9—figure supplement 1E). We compared Syt-7 KO cells to WT cells and to Syt-7 KO cells overexpressing Syt-7 (i.e. rescue experiments) using lentiviral transduction. If there is interaction between Syt-7 and Syt-1, it likely involves the different $Ca^{2+}$-affinities: Syt-7 binds to $Ca^{2+}$ with half-maximal binding around or

below 1 μM (0.3–2 μM), whereas Syt-1 binds $Ca^{2+}$ at higher concentrations (10–150 μM) (*Bhalla et al., 2005*; *Sugita et al., 2002*). Interaction between the isoforms should be revealed when increasing the $[Ca^{2+}]_i$ in two steps: first, an increase from resting levels to an intermediate value (below 1 μM); second, a rapid increase to higher values (>15 μM). This should allow a sizeable fraction of the Syt-7 molecules to bind $Ca^{2+}$ at the intermediate step, whereas Syt-1 would not bind until after the second stimulus. This can be achieved by accurate adjustment of the $[Ca^{2+}]_i$ using caged-$Ca^{2+}$ (nitrophenyl-EGTA) infused through a patch-pipette, and a monochromator Xenon lamp oscillating between 350 and 380 nm, which slowly uncages $Ca^{2+}$, thereby raising the pre-stimulation $[Ca^{2+}]_i$ while simultaneously measuring it by fura dyes (*Voets, 2000*), to allow on-line control. After spending ~20 s at the intermediate calcium concentration, a rapid uncaging stimulus was delivered by a UV flash lamp; the resulting secretion constitutes the output of the experiment and is shown in *Figure 2*.

We first performed uncaging experiments from a relatively low 'pre-stimulation' $[Ca^{2+}]$ (*Figure 2A–E*) of around 180 nM. Under these conditions, knockout of Syt-7 caused a decrease in secretion as measured by both capacitance and amperometry (*Figure 2A,E*; *Figure 2—figure supplement 1A*), whereas re-expression of Syt-7 caused rescue of secretion to WT levels (*Figure 2A,E*). Kinetic analysis identified (statistically non-significant) reductions in the average RRP and SRP size in the absence of Syt-7, whereas rescue by Syt-7 overexpression generated an even larger SRP than that observed for endogenous Syt-7 levels (*Figure 2B*). Syt-7 expression did not significantly affect the kinetics of slow or fast release (*Figure 2C*). The sustained component of release was significantly reduced by the absence of Syt-7 and rescued upon re-expression (*Figure 2D*). Expression analysis by immunofluorescence showed that lentiviral expression of the WT Syt-7 results in levels ~ twofold higher than endogenous Syt-7 (*Figure 2—figure supplement 2A–B*).

We next raised the prestimulation $[Ca^{2+}]$ to ~0.8 μM, which is around the concentration that induces maximal priming in chromaffin cells (*Voets, 2000*); above this concentration, the primed pools are partly emptied due to ongoing fusion. Under these conditions, Syt-7 expressing cells displayed markedly more burst-like secretion (*Figure 2F,J*), consistent with $Ca^{2+}$-dependent vesicle priming. Both fast (RRP) and slow (SRP) burst secretion were strongly and significantly decreased by Syt-7 knockout and rescued by Syt-7 expression (*Figure 2G*). The kinetics of fusion from the RRP and SRP were sped up by Syt-7 knockout, and this was also rescued upon reexpression (*Figure 2H*). Curiously, the sustained rate was reduced in the Syt-7 KO, but this aspect was not rescued upon overexpression (*Figure 2I*). However, RRP and SRP sizes in the KO + Syt-7 condition were even higher than in Syt-7 WT cells, leading to an overshoot in overall secretion within the first second after stimulation (*Figure 2F,G*). Therefore, the lack of rescue of the sustained component might be caused by the twofold overexpression of Syt-7, which in the presence of raised pre-stimulation $[Ca^{2+}]$ stimulates priming of vesicles beyond WT levels, moving vesicles that normally prime and fuse during the sustained component into the RRP and SRP. Recently, it was reported that kiss-and-run fusion events are upregulated in the Syt-7 KO (*Zhang et al., 2019*), but see *Segovia et al., 2010*. Kiss-and-run fusion would cause catecholamine release without net capacitance change. We therefore performed amperometric measurements in parallel with capacitance measurements; these measurements cannot distinguish between RRP and SRP fusion, due to the diffusional delay before catecholamines are recorded at the amperometric fiber, but importantly the total measured amperometric charge and capacitance changed in parallel (*Figure 2A,F* bottom panels; quantification of total amperometric release in *Figure 2—figure supplement 1*), indicating that vesicle fusion and adrenaline release were similarly affected by the presence of Syt-7.

Almost all fast secretion from the RRP in chromaffin cells depends on Syt-1 (*Nagy et al., 2006*; *Voets et al., 2001*) (see also *Figure 1*). Therefore, the data showing potentiation of the RRP size by Syt-7 indicates a cooperative/competitive interplay between Syt-7 and Syt-1, such that Syt-7 increases the amplitude of Syt-1 driven exocytosis (cooperation, *Figure 2G*), but slows down its fusion kinetics (competition, *Figure 2H*), at least under conditions where prestimulation $[Ca^{2+}]$ is relatively high.

The ability of Syt-7 to increase the release burst size indicates a role in vesicle priming. The main $Ca^{2+}$-dependent priming step is in chromaffin cells located upstream of the SRP, which in turn is connected to the RRP (*Voets, 2000*). By acting at the upstream priming step ('priming I', *Houy et al., 2017*), Syt-7 regulates both SRP and RRP size. In order to measure $Ca^{2+}$-dependent priming in the presence and absence of Syt-7, we varied pre-stimulation $[Ca^{2+}]$, followed by an uncaging flash to

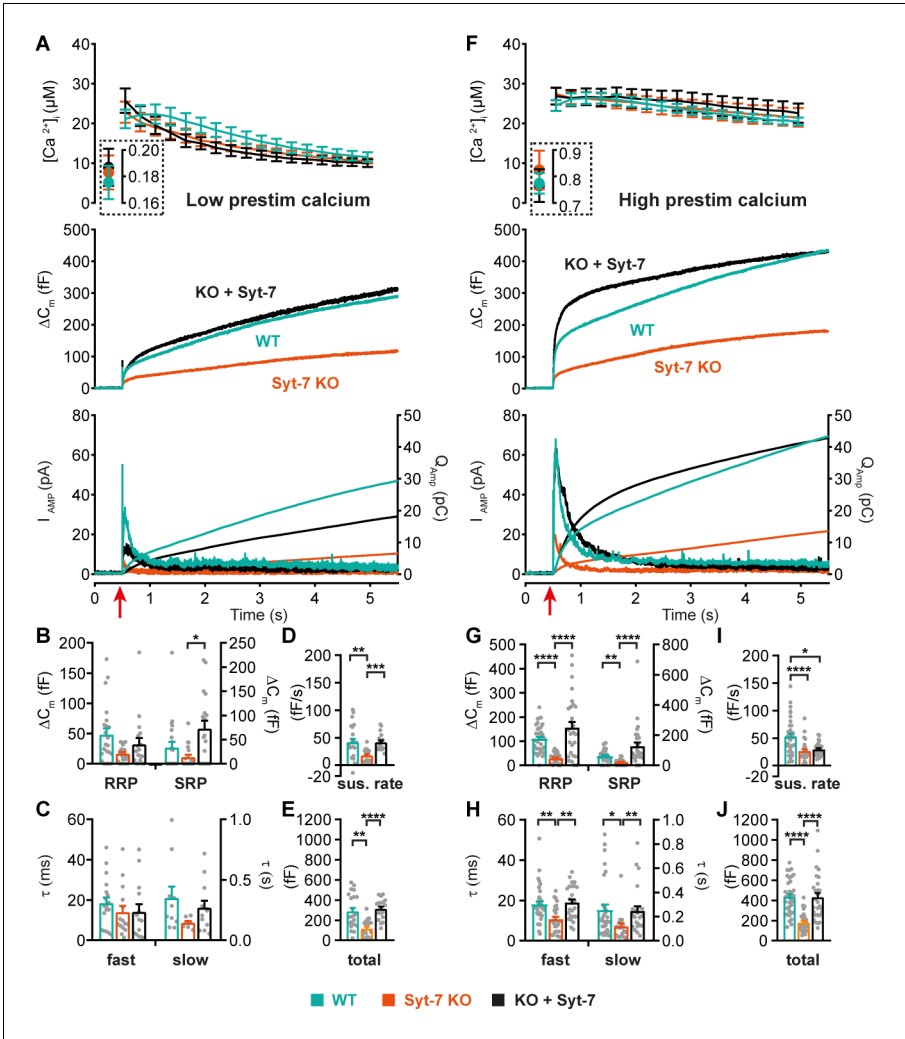

**Figure 2.** Syt-7 potentiates primed vesicle pool sizes at higher prestimulation $[Ca^{2+}]$. (A) Calcium uncaging experiment from low prestimulation $[Ca^{2+}]$ in WT cells (persian green), Syt-7 KO cells (vermilion) and in Syt-7 KO cells overexpressing Syt-7 (black traces). Panels are arranged as in *Figure 1A*. (B) Sizes of the RRP and SRP. (C) Time constants of fusion for fast (i.e. RRP) and slow (i.e. SRP) secretion. (D) Sustained rates of secretion. (E) Total capacitance increase. (F) Calcium uncaging experiment from high prestimulation $[Ca^{2+}]$ in WT cells (green), Syt-7 KO cells (vermilion) and in Syt-7 KO cells overexpressing Syt-7 (black traces). Panels arranged as in *Figure 1A*. (G) Sizes of the RRP and SRP. (H) Time constants of fusion for fast (i.e. RRP) and slow (i.e. SRP) secretion. (I) Sustained rates of secretion. (J) Total capacitance increase. When stimulated from high prestimulation $[Ca^{2+}]$, Syt-7 expression potentiated RRP and SRP size. Data information: In (A–J) data with error bars are presented as mean ± SEM; in (A and F), the traces are the mean of all cells. *p<0.05; **p<0.01; ***p<0.001; ****p<0.0001. Kruskal-Wallis test with Dunn's post-hoc test. Number of cells in (A–E): Syt-7 WT: N = 22 cells; Syt-7 KO: N = 19 cells; Syt-7 KO + Syt-7: N = 18 cells, in (F–J) Syt-7 WT: N = 36 cells; Syt-7 KO: N = 27 cells; Syt-7 KO + Syt-7: N = 28 cells. Note that in cases where a cell did not have a given pool (SRP or RRP), the size of that pool was set to zero, and no time constant was estimated.

The online version of this article includes the following figure supplement(s) for figure 2:

**Figure supplement 1.** Amperometric charge quantification.

**Figure supplement 2.** Mutation of $Ca^{2+}$-binding sites in Syt-7 abolishes rescue function.

assess the secretory burst (i.e. secretion within 0.5 s after the flash, which approximately includes the RRP and the SRP). In WT cells, preflash $[Ca^{2+}]$ up to approximately 0.5 μM triggered $Ca^{2+}$-dependent priming and a corresponding increase in burst release; at higher calcium concentrations the burst size decreases because the high prestimulation calcium causes some release before the flash

(*Figure 3*; *Voets, 2000*), thereby reducing the releasable pool sizes. Strikingly, $Ca^{2+}$-dependent priming was almost absent in Syt-7 KO cells (*Figure 3*). In contrast, expression of Syt-7 in KO cells enhanced priming beyond WT levels at $[Ca^{2+}]$ above 0.2 µM (*Figure 3*). Although $Ca^{2+}$-dependent priming was barely detectable in the Syt-7 KO in the overall titration (*Figure 3*), we noticed that the burst size was slightly higher in the Syt-7 KO in the two-group comparison (*Figure 2A* vs *2F*). Further analysis of this showed that the RRP was significantly increased in the Syt-7 KO (low prestimulation $[Ca^{2+}]$, RRP size: 16.7 ± 3.10 fF; high prestimulation $[Ca^{2+}]$, RRP size: 29.1 ± 4.08 fF. p=0.0480, Mann Whitney test), whereas the increase in SRP was not significant (low prestimulation $[Ca^{2+}]$, SRP size: 13.3 ± 5.35 fF; high prestimulation $[Ca^{2+}]$, SRP size: 20.7 ± 3.84 fF. p=0.0915, Mann Whitney test). Thus, Syt-7 is necessary for strong $Ca^{2+}$-dependent priming in adrenal chromaffin cells, and overexpressing the protein further boosts this function; in Syt-7 KO cells, a small $Ca^{2+}$-dependent priming effect persists.

We next investigated the role of $Ca^{2+}$-binding to Syt-7 for its function in chromaffin cells. We mutated the 'top loops' of either one or both C2-domains to eliminate $Ca^{2+}$-binding, using previously characterized substitutions of the $Ca^{2+}$ coordinating aspartate residues (*Bacaj et al., 2015*). Expression analysis by immunofluorescence (*Figure 2—figure supplement 2A–B*) showed that the construct mutated in the C2B domain (C2B*) was expressed at similar levels (~twofold) as the WT Syt-7 construct. However, constructs mutated in the C2A-domain (C2A*), or both in the C2A and C2B-domain (C2AB*) were expressed at lower levels, comparable to endogenous Syt-7 levels in WT cells (*Figure 2—figure supplement 2A–B*). Overexpressed protein was localized to vesicles, but some of the protein also accumulated in larger clusters (*Figure 2—figure supplement 2A*). Nevertheless, rescue data shown above (*Figures 1* and *2*) demonstrated that our WT construct is functional. Using $Ca^{2+}$ uncaging we found that all three mutations (C2A*, C2B*, C2AB*) failed to rescue the increase in RRP and SRP size (*Figure 2—figure supplement 2C–F*). Rescue by WT Syt-7 was confirmed in parallel experiments. These findings deviate from results in neurons indicating that only $Ca^{2+}$-binding to the C2A-domain is required for Syt-7 function (*Bacaj et al., 2015*), which was attributed to a higher membrane-affinity of the C2A-domain (*Voleti et al., 2017*). However, our findings agree with a recent report that the C2 domains of Syt-7 act synergistically in binding to PI(4,5)P$_2$-containing membranes (*Tran et al., 2019*), and previous data obtained from knockin (C2B-mutated) Syt-7 mice, which displayed a phenotype similar to Syt-7 KO in chromaffin cells (*Schonn et al., 2008*).

## Syt-7 reduces the depriming rate and increases the priming rate

Above, we showed that Syt-7 is involved in LDCV priming. Priming is a reversible process (*Hay and Martin, 1992*; *Smith et al., 1998*), described by a (forward) priming rate constant $k_1$, and a (backward) depriming rate constant, $k_{-1}$ (*Figure 4A*). In a simple one-pool model (without release sites, see below), the forward priming rate is [Depot] • $k_1$, where [Depot] is the size of the upstream Depot pool. The

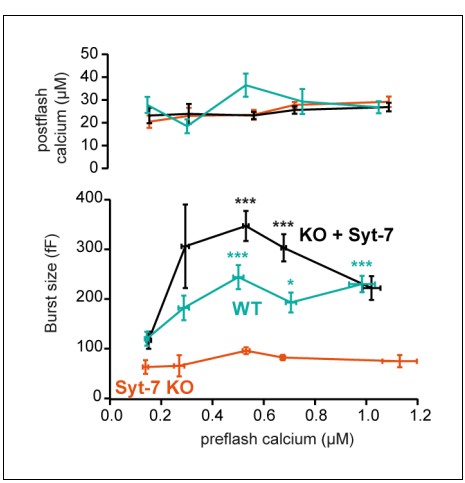

**Figure 3.** Calcium-dependent steady-state priming depends on Syt-7 expression. Titration of burst of secretion (i.e. secretion within the first 0.5 s of the uncaging flash, approximately corresponding to the fusion of the RRP and SRP) against pre-stimulation $[Ca^{2+}]$. Top panel: post-stimulation $[Ca^{2+}]$, bottom panel: Burst size against pre-stimulation $[Ca^{2+}]$. Syt-7 KO cells (vermilion) displayed no strong dependence on $[Ca^{2+}]$. Calcium-dependent priming was strong in cells overexpressing Syt-7 (black), and intermediate in WT cells expressing Syt-7 at endogenous levels (persian green). Data information: Data are presented as mean ± SEM. *: p<0.05; ***p<0.001. Testing was by Kruskal-Wallis test with Dunn's post-hoc test. Cells were pooled in 0.2 µM $[Ca^{2+}]$ bins from 0.0 µM to 0.8 µM and a final bin 0.8–1.2 µM for a total of 5 bins. Statistical testing is relative to the burst size at the lowest $[Ca^{2+}]$ bin for the same genotype. The number of cells in each bin from low to high $[Ca^{2+}]$: WT: N = 29, 9, 9, 10, 22 cells; Syt-7 KO: N = 10, 12, 16, 43, 15 cells; Syt-7 KO + Syt-7: N = 12, 10, 28, 28, 20 cells.

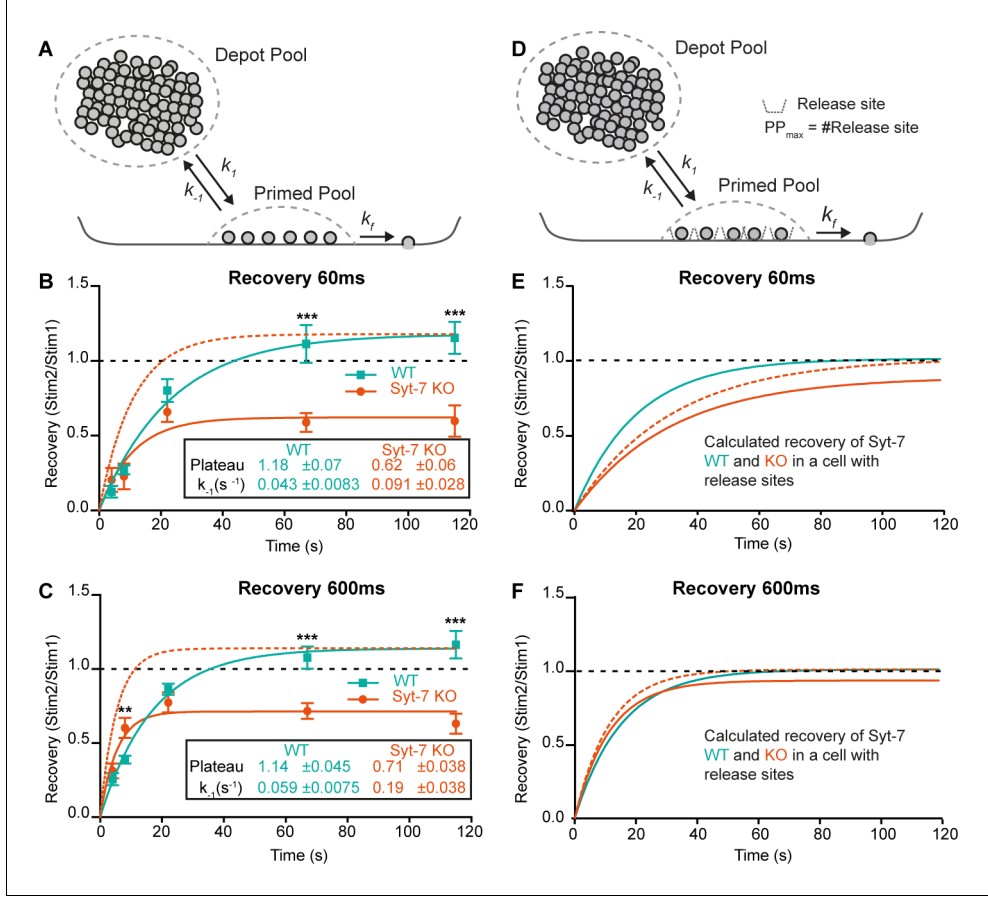

**Figure 4.** Syt-7 increases forward priming and decreases depriming. (A) Simple model (Model I) featuring a single primed vesicle pool, a reversible priming reaction (forward rate: $k_1$; reverse rate: $k_{-1}$), and a fusion rate $k_f$. (B) Recovery in WT cells (persian green) and Syt-7 KO cells (vermilion) of secretion at 60 ms after an uncaging stimulus, approximately corresponding to the RRP. Stim1, Stim2 = amplitude of secretion after first, or second, stimulus. Shown are mean ± SEM, plus a fit of a mono-exponential recovery curve (lines). The fit to the Syt-7 KO is also shown after scaling to the same amplitude as the WT curve (vermilion broken line), to show the faster kinetics. The fitted parameters: Plateau and the rate constant of recovery, which is the rate constant for depriming, $k_{-1}$, under simplified assumptions (see text). Both Plateau and $k_{-1}$ are significantly different from each other (Extra sum-of-squares F test for comparison of models, p<0.0001). (C) Same as B, but secretion at 600 ms after uncaging was used, approximately corresponding to the fusion of both RRP and SRP. (D) Model (*Model II*) featuring a single primed vesicle pool, limited by a fixed number of release sites, $PP_{max}$, a reversible priming reaction (forward rate: $k_1$; reverse rate: $k_{-1}$), and a fusion rate $k_f$. (E) Recovery (at 60 ms) in the WT (green, *Model II*) with parameters recalculated from the fit of *Model I* to the data (panel B, see Materials and methods), and Syt-7 KO curve (vermilion), with the same change in priming and depriming rate as observed experimentally, now translated to a release site model, and after scaling to the WT amplitude (vermilion broken line). Under these circumstances, recovery in the Syt-7 KO trails the WT. (F) Recovery (at 600 ms) in the WT (*Model II*) with parameters recalculated from panel C, and in the KO (vermilion), when introducing the same changes as observed experimentally, translated to a release site model, and after scaling to the WT amplitude (vermilion broken lines). Under these circumstances, the Syt-7 KO leads the WT trace, but the differences are small. Data information: Data are presented as mean ± SEM. **: p<0.01; ***: p<0.001. In (B, C) Student's t-test: test between genotypes (WT vs. KO) at the same inter stimulus intervals; Mann Whitney test: (600 ms: WT22s vs. KO22s).

The online version of this article includes the following figure supplement(s) for figure 4:

**Figure supplement 1.** Differences in unpriming rate can be distinguished during pool recovery, if the Primed Pool is not limited by release sites.

**Figure supplement 2.** Double stimulation experiments in Syt-7 WT and KO cells.

depriming rate ($k_{-1}$) has no contribution to refilling for an empty RRP (refilling is basically linear as it starts out), but with a growing RRP the effect of $k_{-1}$ causes a slowing down, resulting in an exponential recovery time course with time constant $1/k_{-1}$ (*Equation 10*, Materials and methods, *Figure 4—figure supplement 1*):

$$\mathrm{RRP}(t) = \frac{k_1 \cdot [\mathrm{Depot}]}{k_{-1}} \left(1 - e^{-k_{-1} \cdot t}\right)$$

To distinguish between effects of Syt-7 on priming ([Depot] • $k_1$) and depriming ($k_{-1}$), we applied two sequential uncaging flashes separated by different inter-stimulus intervals in Syt-7 KO and WT cells (*Figure 4—figure supplement 2A–C*). We kept the prestimulation [$Ca^{2+}$] relatively low (250–350 nM). Due to the small amplitudes at short recovery intervals, fitting of exponentials was not reliable. Instead, we used the capacitance increase at 60 ms after the flash (approximately corresponding to fusion of the RRP), and at 600 ms (corresponding to the fusion of both RRP and SRP) to assess recovery. Recovery was clearly visible in mean capacitance traces in both the Syt-7 KO and the WT (*Figure 4—figure supplement 2A–C*). Plotting the recovery curves (*Figure 4B–C*), they displayed a slight overshoot in the WT case, but remained incomplete in the Syt-7 KO, even after 120 s. Fitting single-exponential recovery curves allowed us to directly estimate the depriming rate, $k_{-1}$ (see above). This rate was significantly higher in the Syt-7 KO whether the 60 ms or the 600 ms time point was considered (*Figure 4B–C*, *Table 1*). Together with the different steady-state pool sizes, this allows us to calculate the forward priming rate, i.e. [Depot] • $k_1$, which was similar for Syt-7 KO and WT at 60 ms (WT: 3.21 ± 0.63 fF/s, Syt-7 KO: 3.07 ± 0.95 fF/s; *Table 1*) and even higher for the Syt-7 KO than the WT at the 600 ms time point (WT: 8.60 ± 1.12 fF/s, Syt-7 KO: 15.8 ± 3.32 fF/s; *Table 1*). The calculated numbers are the priming rates before the first stimulation, because they derive from the pool sizes measured at the first stimulation. We interpret the slight overfilling (in the WT) or the incomplete recovery (in the Syt-7 KO) as originating from post-stimulation changes in the forward priming rate. Accordingly, in the WT the priming rate increases slightly after stimulation; however, in the Syt-7 KO the priming rate drops following stimulation (WT at 60 ms: 3.21 ± 0.63 fF/s before and 3.78 ± 0.78 fF/s after stimulation, Syt-7 KO at 60 ms: 3.07 ± 0.95 fF/s before and 1.90 ± 0.62 fF/s after secretion, *Table 1*). Given that the priming rate is [Depot] • $k_1$, and $Ca^{2+}$-uncaging likely leads to some reduction in [Depot], a compensatory increase in $k_1$ is necessary to keep the priming rate constant from one stimulation to the next, and an even larger $k_1$ increase is necessary to support overfilling in the WT. We conclude that $k_1$ increases in the Syt-7 WT after stimulation, whereas in the Syt-7 KO this increase is absent or insufficient, leading to incomplete recovery.

Previously, slowed recovery kinetics in the Syt-7 KO was reported in mammalian synapses (*Chen et al., 2017*; *Liu et al., 2014*) (but see *Guan et al., 2020*, for different finding in the *Drosophila* neuromuscular junction). In chromaffin cells, we found incomplete, but kinetically faster, recovery kinetics (*Figure 4B–C*); we next considered how these two observations might be reconciled. In chemical synapses, priming relies on distinct release sites, which limit the size of the RRP. Specialized release sites are missing in chromaffin cells, where the RRP (and SRP) are free to change size when

**Table 1.** Secretion parameters for Syt-7 WT and KO.

Estimated parameters for Syt-7 WT and KO when secretion is measured 60 ms or 600 ms after $Ca^{2+}$-uncaging, which corresponds approximately to the fusion of the RRP, or the RRP + SRP, respectively. Syt-7 elimination resulted in an increase in the reverse priming rate ($k_{-1}$), and a reduction in forward priming rate ($k_1$*DP, where DP is the size of the Depot Pool, and $k_1$ is the rate constant for priming) following stimulation.

| | 60 ms (approx. RRP) | | 600 ms (approx. RRP+SRP) | |
| --- | --- | --- | --- | --- |
| | Syt-7 WT | Syt-7 KO | Syt-7 WT | Syt-7 KO |
| Pool size (fF) | 74.2 ± 3.3 | 33.6 ± 0.77 | 145 ± 4 | 85.4 ± 2.7 |
| $k_{-1}$ ($s^{-1}$) | 0.043 ± 0.0083 | 0.091 ± 0.028 | 0.059 ± 0.0075 | 0.19 ± 0.038 |
| $k_1$ (Before Stim) (fF/s) | 3.21 ± 0.63 | 3.07 ± 0.95 | 8.60 ± 1.12 | 15.8 ± 3.32 |
| $k_1$ (After Stim) (fF/s) | 3.78 ± 0.78 | 1.90 ± 0.62 | 9.8 ± 1.3 | 11.29 ± 2.44 |

priming rates change – this assumption was implicit in the presentation above (*Model I*, *Figure 4A*). Thus, manipulations that change priming can cause large changes in the RRP and SRP size in chromaffin cells, but not in neurons (an example is the effect of β phorbolesters [*Basu et al., 2007*; *Lou et al., 2005*; *Smith et al., 1998*]).

To investigate the consequences of release sites, we constructed *Model II*, where a fixed number of release sites put an upper limit on the primed vesicle pool size (Materials and methods, *Figure 4—figure supplement 1B*). The release sites were assumed to be 90% occupied at rest and to recycle immediately after use, which yields exponential recovery, making it possible to recalculate all parameters of *Model II* directly from our parameter estimates in *Model I* to yield recovery curves with identical kinetics. In this 'neuronal' model, changing depriming ($k_{-1}$) had much smaller effects on the recovery kinetics (*Figure 4—figure supplement 1B*). Decreasing forward priming ($k_1$) reduced both recovery kinetics and pool size (*Figure 4—figure supplement 1B*), which is different from *Model I* (where $k_1$ does not affect recovery kinetics). We finally asked how the observed changes in depriming and priming rate in the Syt-7 KO would affect recovery if translated directly to a release site model (*Figure 4D*). Using recovery of the 60 ms pool (the RRP), the reduction in forward priming rate following stimulation dominated recovery, resulting in slower recovery in the Syt-7 KO than in the WT (*Figure 4E*). Considering the 600 ms pool (SRP + RRP), recovery in the Syt-7 KO preceded the WT, also after normalization (*Figure 4F*), which is due to the overall higher forward priming rates we estimated for this pool in the Syt-7 KO; however, the recovery curves were overall similar, and probably indistinguishable in the presence of experimental variation.

We conclude that the faster, but incomplete recovery kinetics of the RRP in the Syt-7 KO chromaffin cell translates into slower recovery kinetics in a synapse, which reconciles our findings with data from mammalian synapses (*Chen et al., 2017*; *Liu et al., 2014*). It also demonstrates that an experimental advantage of the chromaffin cell is that effects on (forward) priming and (reverse) depriming rates can be separated – this is very difficult in the presence of a limited number of release sites.

NSF (N-ethylmaleimide sensitive factor) disassembles cis-SNARE-complexes following fusion, but it might also disassemble trans-SNARE-complexes leading to depriming, unless this activity is blocked by Munc18-1 and Munc13-1 (*He et al., 2017*; *Ma et al., 2013*; *Prinslow et al., 2019*). It was found that treatment with N-ethylmaleimide (NEM) stabilized the primed vesicle state in neurons where priming had been rendered labile through deletion of Munc13-1 or replacement of Munc18-1 with Munc18-2 (*He et al., 2017*). The effect was interpreted as the prevention of depriming through inhibition of NSF (N-ethylmaleimide sensitive factor). Since we showed above that Syt-7 regulates the depriming rate, we asked whether NEM could partially substitute for Syt-7. We performed $Ca^{2+}$-uncaging in Syt-7 KO and WT cells from low pre-stimulation $[Ca^{2+}]$ (data on higher prestimulation $[Ca^{2+}]$ below) in the presence or absence of 200 μM NEM, which was included in the pipette solution. Strikingly, this led to an increase in the burst and total amount of release in the Syt-7 KO (*Figure 5A–D*). Indeed, in the presence of NEM, secretion in the Syt-7 KO became indistinguishable from WT, whereas in WT cells, NEM was without significant effect (*Figure 5A–D*). Kinetic analysis showed that NEM specifically increased RRP size (*Figure 5E*), but with no significant effect on SRP size (*Figure 5G*) or on the fast or slow time constants of release. Since no effect of NEM was found in the Syt-7 WT, the effect of NEM is occluded by Syt-7 expression. This finding is consistent with a function of Syt-7 to protect SNARE-complexes from disassembly, which probably underlies the function of Syt-7 to decrease the depriming rate. In the Syt-7 KO, SNARE-complexes are left (partly) unprotected and priming can therefore be rescued by NEM, whereas in the Syt-7 WT, where SNARE-complexes are already protected, NEM has no effect.

Overall, these data show that Syt-7 acts to promote priming calcium-dependently and also to inhibit depriming, likely by preventing SNARE-complex disassembly either directly or indirectly.

## Syt-7 assists in ubMunc13-2/phorbolester-dependent priming

The involvement of Syt-7 in priming prompts the question how this function relates to the canonical Munc13 priming proteins? These proteins contain the essential MUN-domain, which opens up syntaxin-1 within the syntaxin-1:Munc18-1 complex to enable SNARE-complex assembly (*Basu et al., 2005*; *Stevens et al., 2005*; *Yang et al., 2015*), driving vesicle priming (*Sørensen et al., 2006*; *Walter et al., 2010*). The main Munc13-protein in chromaffin cells, ubMunc13-2, acts at the same upstream priming step (priming I) (*Man et al., 2015*), as Syt-7 (see above). Thus, it is important to understand whether functional interrelationships between Syt-7 and Munc13 proteins co-determine

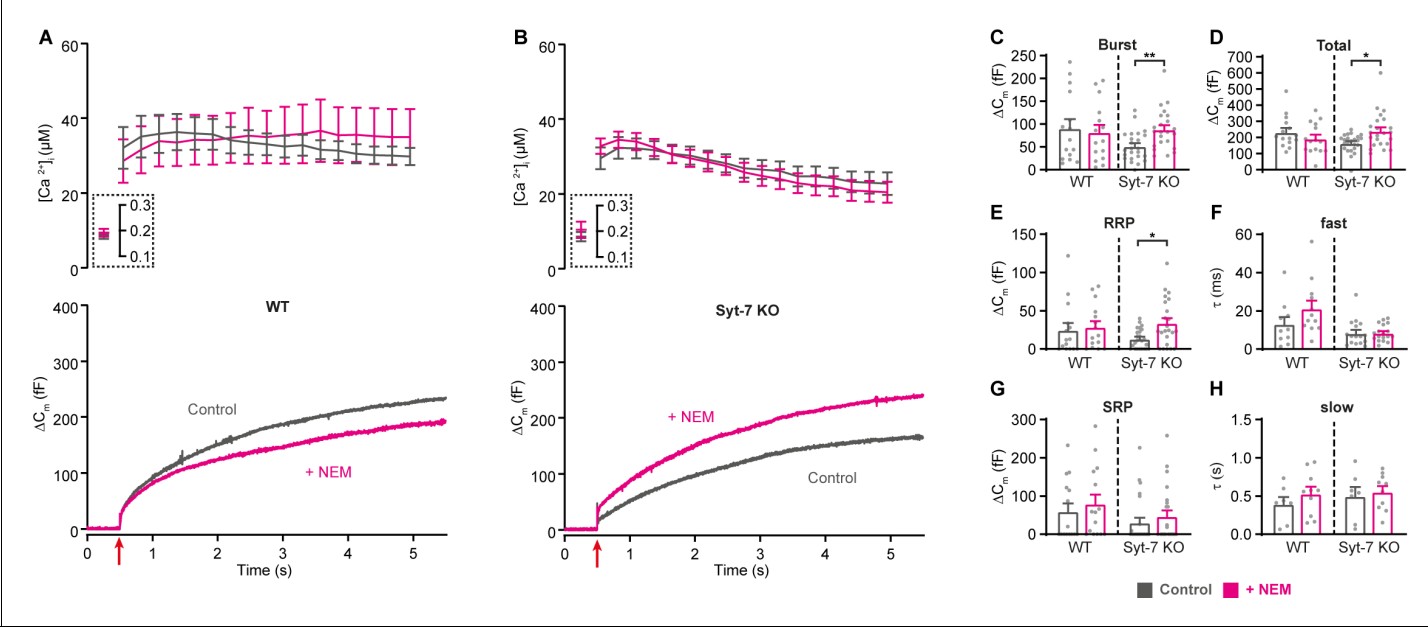

**Figure 5.** Blocking NSF-dependent de-priming occludes the effect of Syt-7 at low prestimulation [Ca$^{2+}$]. (A,B) Calcium uncaging experiment from low prestimulation [Ca$^{2+}$] in WT (A) and in Syt-7 KO (B) control cells (Control, gray) and in cells infused with 200 μM N-Ethylmaleimide (+NEM, magenta). Panels are arranged as in *Figure 1A*, except that amperometric measurements were not included. (C) Size of the burst (secretion within 0.5 s after the flash). (D) Total release (secretion within 5 s after the flash). (E) Size of the RRP. (F) Time constant, τ, of fusion for fast (i.e. RRP). (G) Size of the SRP. (H) Time constant, τ, of fusion for slow (i.e. SRP). Data information: Data are presented as mean ± SEM. *: p<0.05; **: p<0.01, Mann Whitney test comparing control cells to cells infused with NEM from the same genotype. Number of cells: WT control: N = 15 cells; WT + NEM: N = 15 cells; Syt-7 KO control: N = 24 cells; Syt-7 KO + NEM: N = 23 cells.

The online version of this article includes the following figure supplement(s) for figure 5:

**Figure supplement 1.** Blocking NSF had no effect in WT and Syt-7 KO cells when stimulated from high prestimulation [Ca$^{2+}$].

their roles in priming. Munc13-2 overexpression is the strongest known manipulation to increase the primed LDCV pool in chromaffin cells (*Zikich et al., 2008*). Secretion is also potentiated by phorbolesters (*Smith et al., 1998*), which activate Munc13-proteins and Protein Kinase C (*Rhee et al., 2002*; *Wierda et al., 2007*). In chromaffin cells, the effect includes a potentiation of the upstream priming step, leading to a larger exocytotic burst (*Smith et al., 1998*). We investigated whether Syt-7 is involved in phorbolester- and ubMunc13-2-dependent priming.

We first tested the ability of the phorbolester phorbol-12-myristate-13-acetate (PMA, 100 nM) to increase secretion in chromaffin cells (*Figure 6*) while stimulating secretion from low prestimulation [Ca$^{2+}$]$_i$ (<200 nM). Under these circumstances, PMA caused a robust increase in secretion in WT cells, resulting from an increase in SRP and RRP size, but not of the sustained component (overall secretion increased by 126%; *Figure 6A–D*). PMA further led to faster SRP secretion (*Figure 6C*), whereas RRP secretion kinetics were not affected. However, in Syt-7 KO cells, the situation was different: PMA resulted in only a minor potentiation of overall release (21%, *Figure 6E–H*), due to a small increase in RRP (which was statistically significant) and SRP size (p=0.0506; *Figure 6F*). The amperometric measurements were fully in agreement with the capacitance measurements (*Figure 6—figure supplement 1* shows quantification of amperometry). Thus, Syt-7 is necessary for PMA to exert its full priming effect at low prestimulation [Ca$^{2+}$] (data on higher prestimulation [Ca$^{2+}$] are presented below).

Overexpression of ubMunc13-2 in WT chromaffin cells using a Semliki Forest Virus construct resulted in massive secretion, which reached 1.0–1.5 pF (*Figure 7A*; data from Syt-7 WT and KO are the same as in *Figure 2*) when again assayed from a low prestimulation [Ca$^{2+}$] (data on higher prestimulation [Ca$^{2+}$] below). This was due to a massive increase in both RRP and SRP sizes (*Figure 7D,G*), as previously reported from bovine chromaffin cells (*Zikich et al., 2008*). However, we noted that the capacitance curve of the ubMunc13-2 overexpression cells did not have the normal concave

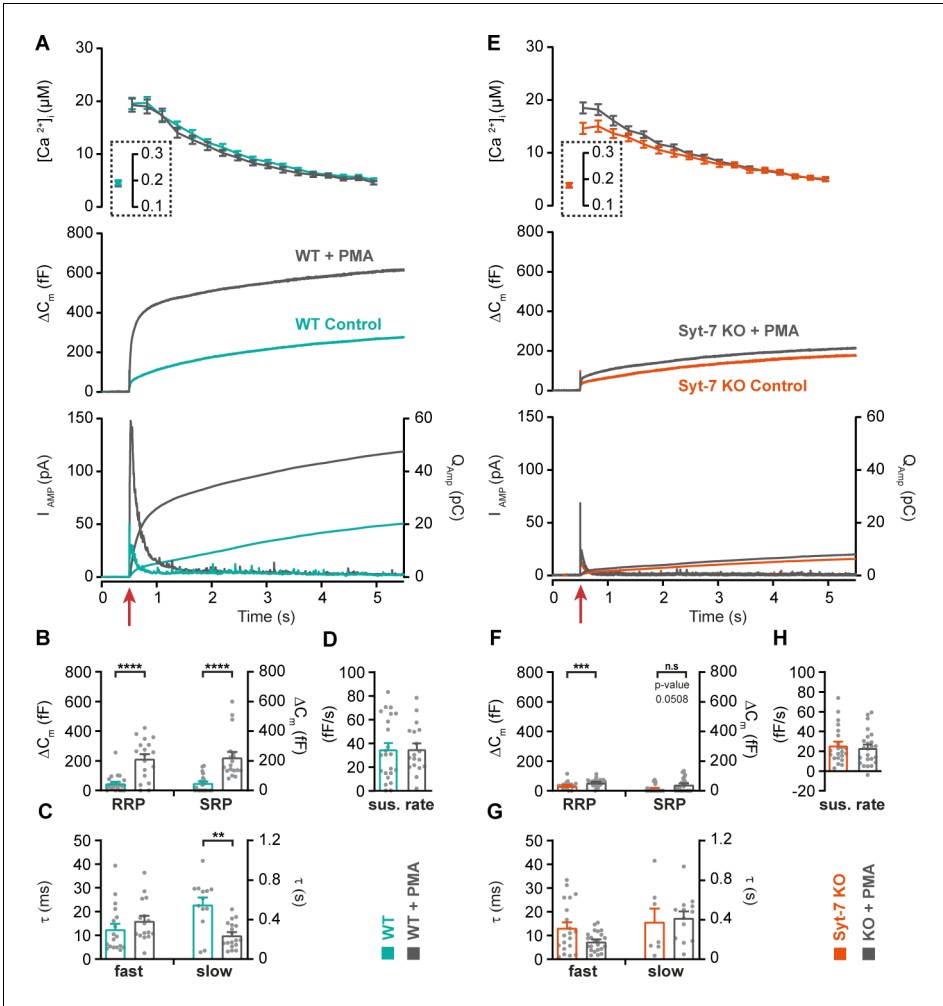

**Figure 6.** Syt-7 stimulates PMA-induced potentiation of release at low prestimulation [$Ca^{2+}$]. (**A**) Calcium uncaging experiment from low prestimulation [$Ca^{2+}$] in Syt-7 WT cells (persian green) and in Syt-7 WT cells perfused with 100 nM phorbol 12-myristate 13-acetate (PMA) (WT + PMA) (gray). Panels are arranged as in *Figure 1A*. PMA treatment strongly augmented the primed pool size in WT cells. (**B**) Sizes of the RRP and SRP. (**C**) Time constants, τ, of fusion for fast (i.e. RRP) and slow (i.e. SRP) secretion. (**D**) Sustained rates of secretion. (**E**) Calcium uncaging experiment from low prestimulation [$Ca^{2+}$] in Syt-7 KO cells (vermilion) and in Syt-7 KO cells perfused with 100 nM PMA (KO + PMA) (gray). PMA-induced potentiation of release was much weaker in Syt-7 KO cells. (**F**) Size of the RRP and SRP. (**G**) Time constants, τ, of fusion for fast (i.e. RRP) and slow (i.e. SRP) secretion. (**H**) Sustained rate of secretion. Data information: Data with error bars (**A–H**) are presented as mean ± SEM; in (**A, E**), the traces are the average of all cells. Statistics: *: p<0.05; **p<0.01; ***p<0.001; ****p<0.0001. Analysis was performed with Mann-Whitney test. Number of cells: WT: N = 23 cells; WT + PMA: N = 18 cells; Syt-7 KO: N = 22 cells; Syt-7 KO + PMA = 24 cells.

The online version of this article includes the following figure supplement(s) for figure 6:

**Figure supplement 1.** Integrated amperometry (mean ± SEM) of WT, WT cells treated with 100 nM phorbol 12-myristate 13-acetate (PMA) (WT + PMA), Syt-7 KO and Syt-7 KO with 100 nM PMA (Syt-7 KO + PMA) stimulated from low prestimulation [$Ca^{2+}$].

**Figure supplement 2.** Application of phorbol esters to Syt-7 WT and KO at higher prestimulation [$Ca^{2+}$].

**Figure supplement 3.** Integrated amperometry (mean ± SEM) of WT, WT cells perfused with 100 nM phorbol 12-myristate 13-acetate (PMA) (WT + PMA), Syt-7 KO and Syt-7 KO treated with 100 nM PMA (syt-7 KO + PMA) stimulated from high prestimulation [$Ca^{2+}$].

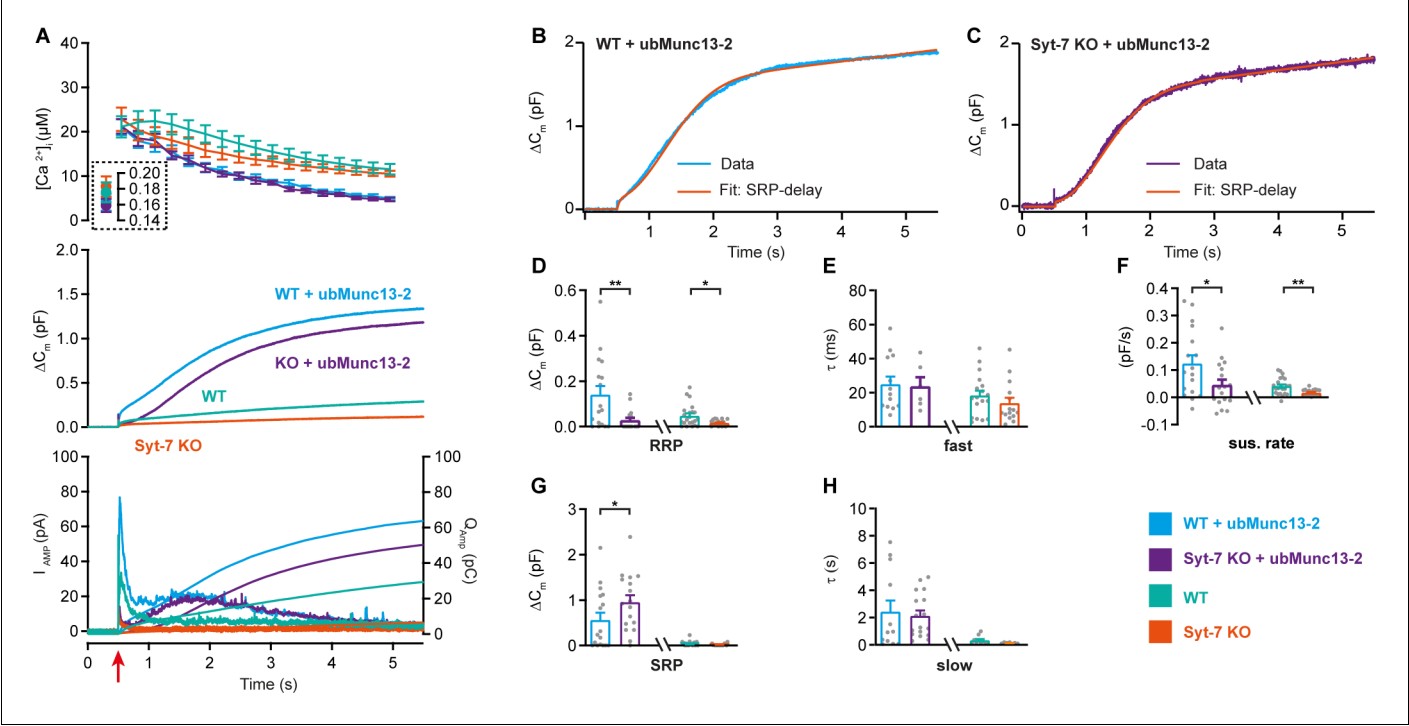

**Figure 7.** Syt-7 stimulates ubMunc13-2-dependent priming at low prestimulation [Ca²⁺]. (A) Calcium uncaging experiment in WT overexpressing ubMunc13-2 (WT + ubMunc13-2) (cyan traces), and Syt-7 KO overexpressing ubMunc13-2 (KO + ubMunc13-2) (purple traces) stimulated from a low prestimulation [Ca²⁺]. Data from Syt-7 WT and Syt-7 KO are the same as in *Figure 2A*. Panels are arranged as in *Figure 1A*. The overexpression of ubMunc13-2 potentiated the release in Syt-7 KO cells after a remarkable delay of the SRP. (B) An example capacitance trace ('Data') from a WT cell overexpressing ubMunc13-2 (cyan trace) with a function taking into account the SRP-delay ('Fit', red trace). (C) An example capacitance trace ('Data') from a Syt-7 KO cell overexpressing ubMunc13-2 ('Fit', purple trace) fitted with a function taking into account the SRP-delay (red trace). (D–H) In the Syt-7 WT + ubMunc13-2 or Syt-7 KO + ubMunc13-2, for each cell recorded, the chi-square values between fit and data were used to judge whether the standard sum of two exponentials and a line function or the function including the SRP delay (see Materials and methods) fitted the traces better. Values from the best fit were averaged to obtain the RRP, SRP sizes and time constants. (D, G) Sizes of the RRP and SRP. (E, H) Time constant, τ, of fusion for fast (i.e. RRP) and slow (i.e. SRP) secretion. Note that the τ for the SRP in the Syt-7 WT + ubMunc13-2 and Syt-7 KO + ubMunc13-2 groups include both the secretory delay and the fusion kinetics (*Equation 7*). Due to the slower τ for the SRP in this data set, we assumed that a τ originated from the RRP if τ ≤60 ms and from the SRP when 60 ms ≤τ ≤1600 ms (se Materials and methods). (F) Sustained rate of secretion. Note that in some cases when fitting with the function taking into account the SRP-delay, a negative sustained rate resulted from the fit. Data information: In (A–F) data with error bars are presented as mean ± SEM; in (A), the traces are the mean of all cells. WT (persian green) and Syt-7 KO (vermilion) were not obtained in parallel experiments, but are displayed here only to illustrate the increase upon ubMunc13-2 overexpression; statistical tests are only conducted for Syt-7 KO + ubMunc13-2 vs Syt-7 WT + ubMunc13-2 and Syt-7 WT vs Syt-7 KO. Note that the RRP size of WT vs Syt-7 KO is significantly different, due to the two-group comparison, which was not the case in *Figure 2A* (three-group comparison). Statistics: *: p<0.05; **: p<0.01, Mann Whitney test. Number of cells: WT + ubMunc13-2: N = 17; Syt-7 KO + ubMunc13-2: N = 17 cells; WT: N = 22; Syt-7 KO: N = 19.

The online version of this article includes the following figure supplement(s) for figure 7:

**Figure supplement 1.** Overexpressing ubMunc13-2 in Syt-7 WT and KO cells at higher prestimulation [Ca²⁺].

form; instead, the curves showed signs of a secondary acceleration, giving rise to a convex curve - this was especially clear in the case of ubMunc13-2 expressed in Syt-7 KO cells (*Figure 7A–C*). This change in secretory kinetics was also observed in the parallel amperometric measurements (*Figure 7A* bottom), where the amperometric current increased again after the first phase, reaching a second maximum around 1.5 s. An inspection of individual traces revealed that the SRP was fusing after a longer delay, whereas RRP fusion kinetics were largely unaffected (*Figure 7B–C* shows examples). To investigate whether this is a feasible interpretation, we fitted secretory traces with an alternative function for a SRP fusing after a delay. The function describing SRP fusion is

$$\Delta C_m = SRP\left(1 - e^{-t^2/\tau_{slow,\,delay}}\right)$$

where $\tau_{slow, delay}$, is the product of the time constants for the delay and for slow fusion itself. Due to the long delay, it is not possible to determine fusion kinetics and delay separately (see Materials and methods for derivation of this function). We used the chi-square value between fit and data to determine whether this function or the standard sum of two exponentials and a line (see Materials and methods) fitted the traces better. Syt-7 KO cells overexpressing ubMunc13-2 could only be fitted with the delayed-SRP model (yielding the best fit in 16 of 17 cells), but also Syt-7 WT expressing ubMunc13-2 more often than not had a delayed SRP (best fit in 10 out of 17 cells). This difference was statistically significant (p=0.0391, when tested as a contingency table using Fisher's exact test). This indicates that a delay is more often present in the absence of Syt-7. In *Figure 7A–G*, we compare the results after ubMunc13-2 expression in WT and Syt-7 KO to the data obtained from WT and Syt-7 KO without overexpression (stimulated from a low prestimulation [Ca$^{2+}$], *Figure 2A–E*). Kinetic analyses are presented in *Figure 7D–H* (note that cells from WT and Syt-7 KO were fitted with two exponentials and a linear function, *Equation 1*, and not obtained in parallel with the other data; therefore, these data were not statistically compared to ubMunc13-2 overexpressing cells). These data show a larger ability of ubMunc13-2 to increase the RRP in Syt-7 WT compared to Syt-7 KO (*Figure 7D*), whereas ubMunc13-2 overexpression increased the SRP-size both in the presence and absence of Syt-7 (*Figure 7G*).

Overall, we have shown that under these circumstances (low prestimulation [Ca$^{2+}$]), PMA requires Syt-7 to strongly increase RRP and SRP size, whereas Munc13-2 requires Syt-7 to increase the RRP, but not the SRP size. However, the SRP tends to fuse with an additional delay in the absence of Syt-7.

## Synaptotagmin-7-independent mechanisms partly stimulate priming at high prestimulation [Ca$^{2+}$]

We next repeated the experiments with phorbolester, ubMunc13-2 overexpression, and NEM treatment from higher prestimulation [Ca$^{2+}$]. With phorbolester treatment, the cells would start to secrete at a high rate already around prestimulation [Ca$^{2+}$] of ~500 nM, which is consistent with findings in the Calyx of Held that phorbolester increases the calcium-sensitivity of vesicle fusion (*Lou et al., 2005*). Under these conditions, we therefore adjusted the high prestimulation [Ca$^{2+}$] to around 400 nM (*Figure 6—figure supplement 2A,E*), since otherwise the primed pools would become depleted before we could measure their size. At these prestimulation [Ca$^{2+}$]$_i$, phorbolester led to a strong increase in RRP size in Syt-7 WT cells, no significant changes in SRP and a mild, but not significant, increase in the sustained component (*Figure 6—figure supplement 2A–D*). In Syt-7 KO, the RRP was also potentiated by PMA (*Figure 6—figure supplement 2F*), although the RRP size in the Syt-7 KO remained smaller than in the Syt-7 WT (*Figure 6—figure supplement 2B*). In addition, the sustained component was significantly potentiated by PMA in the Syt-7 KO. Overall, therefore, total secretion in the Syt-7 KO + phorbolester was only slightly reduced compared to Syt-7 WT + phorbolester (see also *Figure 6—figure supplement 3* for amperometric data and compare with *Figure 6—figure supplement 1* showing amperometric data from low prestimulation [Ca$^{2+}$]), but the size of the RRP remained smaller in the absence of Syt-7.

When expressing ubMunc13-2 in Syt-7 KO and Syt-7 WT and stimulating from a higher prestimulation [Ca$^{2+}$], the RRP was clearly increased by ubMunc13-2 in the Syt-7 KO, although still depressed compared to ubMunc13-2 overexpressing Syt-7 WT cells (ubMunc13-2 in Syt-7 WT: 410 ± 83 fF; ubMunc13-2 in Syt-7 KO: 207 ± 39 fF, p=0.04; *Figure 7—figure supplement 1*). Moreover, in ubMunc13-2 expressing WT cells, the two exponentials and a linear function fitted 12 of 13 cells best, and only one cell was fitted better with an SRP-delay. In contrast, in Syt-7 KO cells the function incorporating a SRP-delay fitted 16 of 16 cells better. This difference was statistically significant (p<0.0001, when tested as a contingency table using Fisher's exact test). This demonstrates that fusion of the ubMunc13-2-induced SRP still occurs in the absence of Syt-7, but that it does so with a significant delay compared to Syt-7 WT cells.

Finally, neither Syt-7 KO nor WT cells were affected by NEM at high pre-stimulation [Ca$^{2+}$] around 400–500 nM (*Figure 5—figure supplement 1*), thereby indicating that alternative molecules protect SNARE complexes under these conditions. These data are consistent with two alternative Syt-7 functions, as demonstrated above, namely the promotion of forward priming ($k_1$) and the inhibition of unpriming ($k_{-1}$). Syt-7 might assist in SNARE-complex formation to increase $k_1$ at high prestimulation [Ca$^{2+}$], and then protect the formed SNARE-complexes (reducing $k_{-1}$) as the [Ca$^{2+}$] relaxes back to

low values; however, Syt-7 does not subserve both sub-functions simultaneously and alternative factors appear responsible for protecting SNARE-complexes at high [Ca$^{2+}$].

Altogether, these data show that when combining increased prestimulation [Ca$^{2+}$] with additional manipulations to increase priming, some priming is seen in the absence of Syt-7, although the RRP size often remains smaller. These data can be accounted for by a mechanism in which Syt-7 acts in priming upstream of or at the same step as Munc13-2/phorbolester (see Discussion).

## Syt-7 promotes the placement of dense-core vesicles at the plasma membrane

High-pressure freezing (HPF) fixation followed by freeze-substitution for electron microscopy has become an established approach for the analysis of secretory vesicle docking in systems such as hippocampal organotypic slice cultures (*Imig et al., 2014*; *Siksou et al., 2009*), and acute adrenal slices (*Man et al., 2015*). This method, especially when combined with high-resolution 3D electron tomography (3D-ET), allows an accurate assessment of SV and LDCV placement down to a few nanometers of distance to the plasma membrane. The assembled SNARE-bundle can bridge membranes separated by as much as 9–15 nm in the presence of complexin (*Li et al., 2011*). Therefore, high-resolution 3D-ET makes it possible to distinguish the close apposition of vesicle and membrane that follows from the assembly of the priming complex, yielding a morphological read-out of priming (*Imig et al., 2014*; *Siksou et al., 2009*). Previous work showed no difference in placement of LDCVs upon deletion of Munc13-1 and Munc13-2 (*Man et al., 2015*). As Syt-7 partly acts to limit depriming at low resting [Ca$^{2+}$] (above), HPF might offer a possibility of correlating LDCV priming with morphological changes.

Following previously established protocols (*Man et al., 2015*), we combined HPF and freeze-substitution with classical 2D-EM and high-resolution 3D-ET. The reason for including 2D-EM is to quantify the overall vesicle distribution and vesicle number in the entire cell, whereas 3D-ET focuses only on membrane-proximal vesicles. Quantitative analysis of 2D-EM images of Syt-7 WT and KO chromaffin cells (*Figure 8A–D*) revealed no differences in vesicle distribution within 2 μm of the plasma membrane (*Figure 8E–F*), in the number of LDCVs per cell profile (*Figure 8G*), the density of LDCVs per cytoplasm area (*Figure 8H*), or in the fraction of LDCVs present within 40 nm of the membrane ('membrane-proximal' vesicles; *Figure 8I*). Close LDCV-membrane apposition was analyzed using 3D-ET. Focusing exclusively on LDCVs placed less than 100 nm from the membrane, the proportion of vesicles placed closer than 40 nm was not changed in the Syt-7 KO (*Figure 8S*). Vesicles observed to be in physical contact with the plasma membrane in tomographic volumes were considered 'docked' and based on the voxel dimension of reconstructed tomograms these vesicles were placed in the 0–4 nm bin (*Man et al., 2015*). The number of docked vesicles (0–4 nm) and vesicles placed between 4 and 6 nm from the plasma membrane was reduced in the Syt-7 KO (*Figure 8P*); only the latter difference was statistically significant. When pooled into a single bin, the number of vesicles placed closer than 6 nm from the plasma membrane was significantly reduced in the Syt-7 KO (*Figure 8T*, p=0.018). The lack of significance in the docked bin (0–4 nm) might be due to a large number of 'dead-end' docked vesicles (*Hugo et al., 2013*; *Verhage and Sørensen, 2008*). Interestingly, the reduction of LDCVs in the vicinity of the membrane in Syt-7 KO cells was accompanied by an significantly increased number of vesicles at slightly larger distances (20–40 nm, insert in *Figure 8P*). We also noted a tendency for vesicles to be of slightly smaller diameter in the Syt-7 KO (*Figure 8R*). Quantitative considerations (Materials and methods) combining the overall density of vesicles (as observed in 2D images) with the accurate determination of vesicle diameter (as obtained in 3D-ET) made it possible to estimate the total number of vesicles in Syt-7 WT cells (~12,334,334 vesicles) and in Syt-7 KO cells (~13,270,270 vesicles), whereas the total number of vesicles that are attached to the plasma membrane (bin 0–4 nm) was ~329 vesicles/cell for Syt-7 WT and ~266 vesicles/cell for Syt-7 KO.

Overall, in the absence of Syt-7 fewer vesicles are placed very near the plasma membrane, and more vesicles become placed at distances (20–40 nm) which are probably beyond the formation range of the priming-complex. This phenotype correlates with the ability of Syt-7 to stimulate priming.

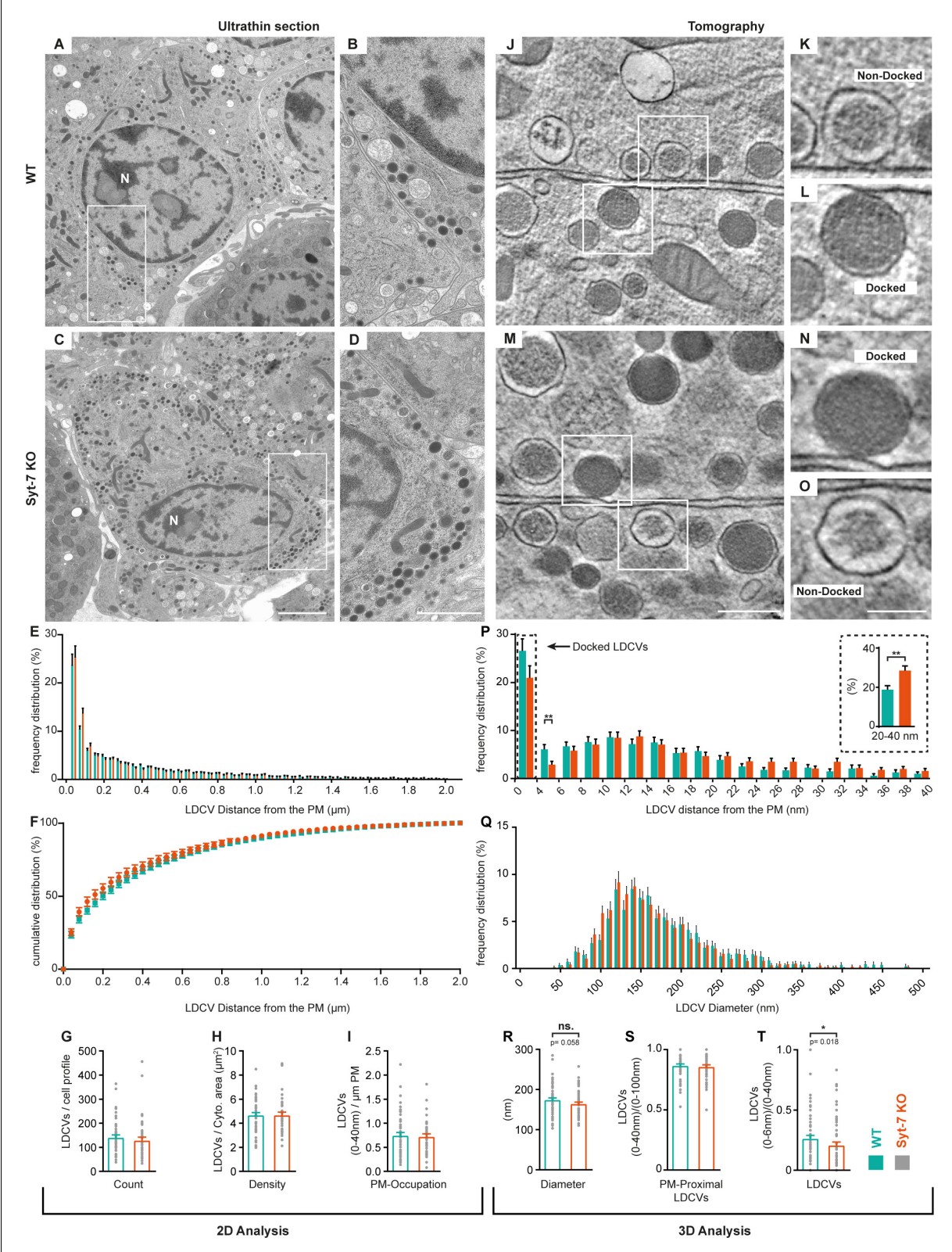

**Figure 8.** Syt-7 induces membrane-apposition of LDCVs. (**A, C**) 2D-EM micrographs of ultrathin adrenal sections from WT (**A**) and Syt-7 KO (**C**) newborn mice. Nucleus is designated (**N**). Scale bar: 2 μm. (**B, D**) Magnification of selection in (**A**) and (**C**), respectively. Scale bar: 1 μm. (**E**) Frequency distribution of large dense core vesicles (LDCVs) within 2 μm from the plasma membrane (PM) in WT (persian green) and Syt-7 KO (vermilion) cells. (**F**) Cumulative frequency plot of (**E**). (**G**) Total number of LDCV per cell profile. (**H**) Number of LDCVs per cytosolic area (Density = LDCVs/μm²). (**I**) Number

*Figure 8 continued on next page*

**Figure 8 continued**

of PM-Proximal LDCVs per µm of PM circumference. 2D analysis revealed normal cell morphology and LDCV distribution in the absence of Syt-7. (J, M) 3D-EM reconstructed tomogram subvolume from WT (J) and Syt-7 KO (M) showing two cells with opposing membranes. Scale bar: 300 nm. K, L, N, O Magnifications of selected regions in (J) and (M); showing docked (L, N) and non-docked (K, O) LDCVs. Scale bar: 150 nm. (P) Frequency distribution of PM-proximal LDCVs where docked vesicles are accumulated in the 0–4 nm bin. Insert is a summation of 20–40 nm into single bins. (Q) Frequency distribution of LDCV diameter. (R) Diameter of LDCVs within 100 nm from the PM. (S) PM-Proximal LDCVs (0–40 nm) normalized to (0–100 nm) LDCVs (0–40 nm LDCVs/0–100 nm LDCVs). (T) Vesicles within 6 nm of the membrane (bin 0–6 nm) normalized to PM-Proximal LDCVs (0–6 nm LDCVs/0–40 nm LDCVs). Overall, the 3D analysis showed that LDCVs 0–6 nm from the PM are markedly reduced in the absence of Syt-7. Non-docked LDCVs in the Syt-7 KO accumulated at 20–40 nm from the PM. Data information: Values are mean ± SEM.*: $p<0.05$; **: $p<0.01$. Student's t-test: (G): $p=0.4240$; (H): $p=0.9711$; (I): $p=0.9241$; (Insert in (P): 20–40 nm): $p=0.0025$; (R): $p=0.0581$; (S): $p=0.6127$. Mann Whitney test: (P): 4–6 nm: $p=0.0039$; (T): $p=0.0184$. Number of cells, 2D analysis: (WT) N = 60 cells, (Syt-7 KO) N = 46 cells; 3D analysis: (WT) N = 74 cells, (Syt-7 KO) N = 74 cells.

## Syt-1 and Syt-7 are both found associated with dense-core vesicles

The work performed above identified several conditions under which Syt-7 stimulated the size of the RRP. This was the case upon elevation of intracellular calcium prior to flash photolysis (*Figure 2*), upon stimulation by phorbolester (*Figure 5*), after ubMunc13-2 expression (*Figure 6*), or when treatment by NEM was performed at low prestimulation [Ca$^{2+}$] (*Figure 7*). The RRP fuses fast (time constant typically 10–20 ms at 20 µM Ca$^{2+}$) and depends on Syt-1 expression, as shown in Syt-1 knockout and rescue experiments (*Nagy et al., 2006*; *Voets et al., 2001*), and as demonstrated here by Syt-1 expression in the Syt-1/Syt-7 Double KO cells (*Figure 1*). Upon Syt-7 expression in the Syt1/Syt-7 DKO there was a minor increase in the RRP (*Figure 1*); however, the time constant was increased to 50 ms, which does not correspond to a typical RRP time constant. These data indicate that Syt-7 is, by itself, not able to fuse vesicles with fast kinetics; instead, Syt-1 must fuse these vesicles. Thus, Syt-7 and Syt-1 cooperate (*Walter et al., 2011*), such that Syt-7 builds up a larger RRP, which then fuses with the help of Syt-1.

This conclusion raises the question whether Syt-1 and Syt-7 are localized to the same vesicles as a prerequisite for their cooperation, or whether cooperation takes place in a different way. Previous work emphasized the difference in localization between Syt-1 and −7 in PC12-cells, and in rat and bovine chromaffin cells, since only limited overlap between either endogenously or exogenously expressed Syt-1 and Syt-7 was identified in immunolabeling experiments performed at light and electron microscopic levels (*Matsuoka et al., 2011*; *Rao et al., 2014*; *Wang et al., 2005*; *Zhang et al., 2011*). In a recent study performed in mouse chromaffin cells, only 3–5% overlap between endogenous Syt-1 and Syt-7 was reported (*Bendahmane et al., 2020*).

We performed double immunostaining using a rabbit polyclonal Syt-7 antibody (Synaptic Systems 105173) and a mouse monoclonal Syt-1 antibody (Synaptic Systems 105011; *Figure 9A*). In chromaffin cells from the Syt-7 KO mouse (*Maximov et al., 2008*) staining against Syt-7 was strongly reduced, as expected (*Figure 9B*), although some background staining remained. Staining for Syt-1 was unchanged in the Syt-7 KO (*Figure 9B*; the Syt-1 antibody was previously verified on Syt-1 KO chromaffin cells [*Nagy et al., 2006*]). Some colocalization between Syt-1 and Syt-7 was observed in these stainings (*Figure 9C*), with Manders' coefficients of 0.19 ± 0.02 (Syt-1 fraction in Syt-7) and 0.16 ± 0.02 (Syt-7 fraction in Syt-1). These numbers were reduced to 0.006 ± 0.003 and 0.039 ± 0.009 in the Syt-7 KO. In these stainings, we included a small amount (0.2%) of glutaraldehyde, because we found that this branched aldehyde led to a better morphology of the fixed cell. Sodium Borohydride was used as a quenching agent. To test that glutaraldehyde does not add unspecific fluorescence, we performed stainings without glutaraldehyde, which yielded the same level of background staining (*Figure 9—figure supplement 1A–C*). Another antibody combination - a mouse monoclonal Syt-7 antibody (MABN665, Sigma-Aldrich, previously used by *Barthet et al., 2018*) together with a rabbit polyclonal Syt-1 antibody (W855, a gift from T.C. Südhof, previously verified on Syt-1 KO chromaffin cells [*Kedar et al., 2015*]) yielded similar results (*Figure 9—figure supplement 1D–F*), but lower background staining and slightly higher Manders' coefficients (0.37 ± 0.03 for Syt-1 fraction in Syt-7; 0.24 ± 0.03 for Syt-7 fraction in Syt-1).

3D-SIM has a resolution approximately twofold higher than confocal microscopy, which made it possible to distinguish single objects in both the Syt-1 and the Syt-7 channels in WT cells (*Figure 9D* shows 110 nm thick optical sections). Strikingly, most Syt-1- and Syt-7-positive structures did not display detectable colocalization (*Figure 9D*: ROI 1), except in a few cases (*Figure 9D*, ROI 2). We used

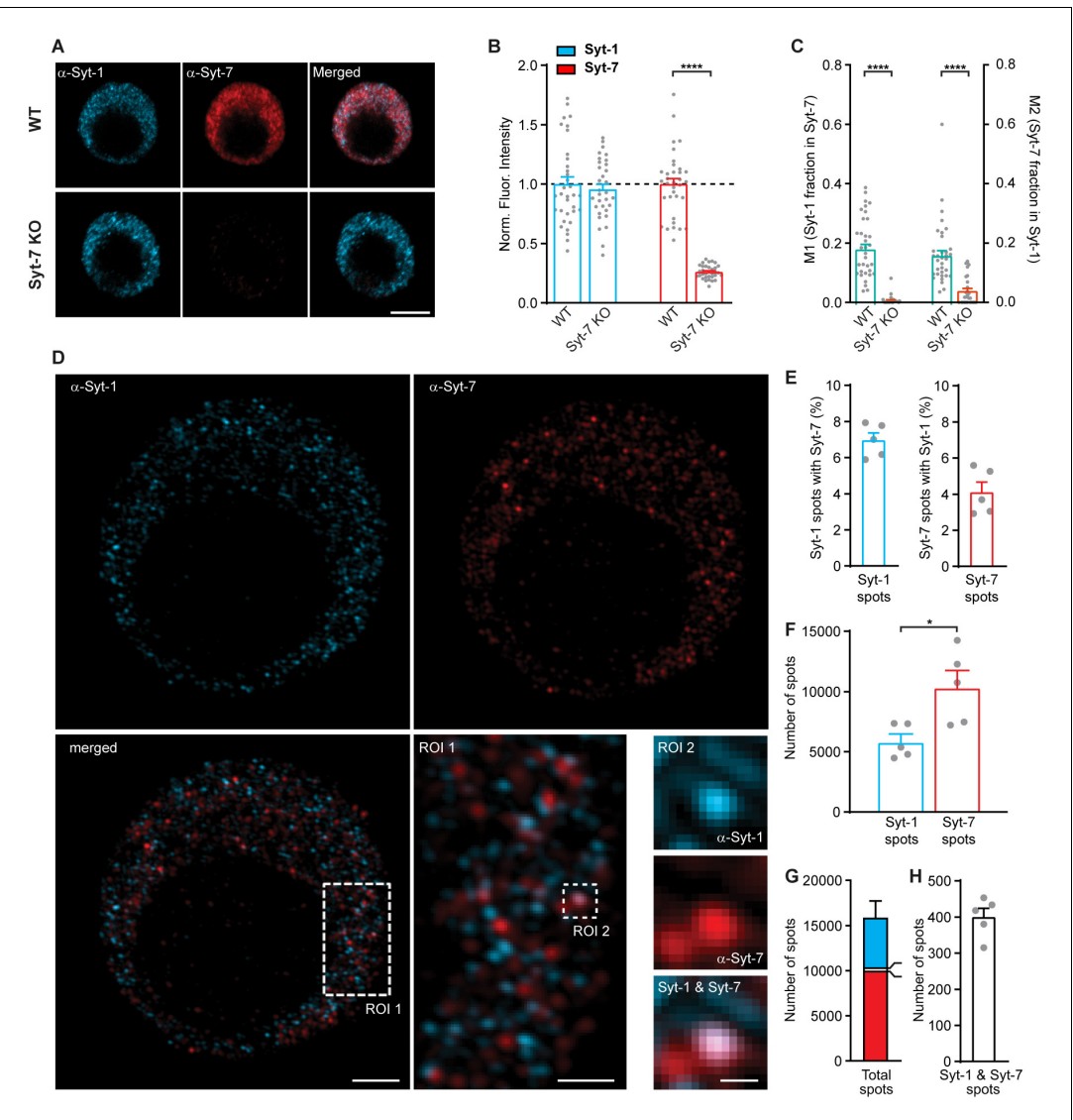

**Figure 9.** Syt-1 and Syt-7 displays limited colocalization. (A) Single confocal slices of new-born mouse chromaffin cells stained against Syt-1 (α-Syt-1) and Syt-7 (α-Syt-7) in WT cells and in Syt-7 KO cells, and merged images. Scale bar: 5 μm. (B) Quantification of staining against Syt-1 and Syt-7 in WT and Syt-7 KO cells. For staining with other antibodies: see *Figure 9—figure supplement 1D–F*. (C) Manders' coefficients M1 and M2 (mean ± SEM) for co-localization analysis of Syt-1 and Syt-7 in WT and Syt-7 KO cells. (D) Single optical slices of new-born WT mouse chromaffin cells stained against Syt-1 (α-Syt-1) and Syt-7 (α-Syt-7) acquired with 3D-structured illumination microscopy (3D-SIM). Scale bar: 2 μm. Bottom right: Magnified ROIs. ROI 1: A section of the cell from the merged-channel image showing that the majority of the spots are identified either as Syt-1 -or Syt-7-positive and few are positive for both isoforms. Scale bar: 1 μm. ROI 2: An example of a spot where Syt-1 and Syt-7 appear co-localized (top panel: α-Syt-1; middle panel: α-Syt-7; bottom panel: A merged image of α-Syt-1 and α-Syt-7 channels). Scale bar: 0.2 μm. (E) Quantification of the percentage of Syt-1 spots that were costained for Syt-7 and vice versa. (F) Number of Syt-1 and Syt-7 spots per cell, while analyzing every third optical slice (thickness of each 0.11 μm) from the top to the bottom of the cells using the ComDet plugin for ImageJ. (G) The total number of spots per cell. Bar colors indicate the proportions of Syt-1 (cyan), Syt-7 (red) and Syt-1/Syt-7 (white) spots. (H) Number of Syt-1/Syt-7 spots where the two isoforms are considered to be colocalized to the same vesicle as shown in ROI two example (D). Data information: Values are mean ± SEM. *: p<0.05; ****: p<0.0001, Student's t-test. Number of cells in (B, C): Syt-7 WT: N = 22 cells; Syt-7 KO: N = 22 cells. Number of cells in (E–H): Syt-7 WT, N = 5 cells.
The online version of this article includes the following figure supplement(s) for figure 9:

**Figure supplement 1.** Syt-1 and Syt-7 limited colocalization.

an automated routine to analyze every third optical section, avoiding detection of the same spots twice. We found that among the Syt-1-positive structures, 7.0 ± 0.4% colocalized with Syt-7, whereas from the Syt-7-positive structures, 4.1 ± 0.6% colocalized with Syt-1 (*Figure 9E–H*). These data are similar to those reported by *Bendahmane et al., 2020*.

In order to understand how Syt-1 and Syt-7 are associated to dense-core vesicles, we performed double stainings against chromogranin A (CgA), a marker of dense-core vesicles, and either Syt-1 or Syt-7. We used the Sigma-Aldrich (MABN665) Syt-7 antibody and the Synaptic Systems (105011) Syt-1 antibody, both of which detect the cytoplasmic domain of the proteins. 3D-SIM revealed that CgA-positive structures were distinct from both Syt-1 and Syt-7 structures (*Figure 10A,E*). Three-dimensional object-based analysis (*Gilles et al., 2017*) showed that the volume of CgA-positive structures (i.e. dense-core vesicles) were significantly larger than Syt-positive structures (*Figure 10C, G*). CgA-positive structures were 0.0077 ± 0.0003 µm$^3$ (mean ± SEM) for the dataset colabeled against Syt-7 (*Figure 10A,C*) or 0.0085 ± 0.0005 µm$^3$ for the dataset colabeled against Syt-1 (*Figure 10E,G*). In contrast, the mean volume of Syt-7 spots (mean ± SEM) was 0.00326 ± 0.00006 µm$^3$ and of Syt-1 spots 0.00256 ± 0.00006 µm$^3$. Furthermore, the number of Syt-7 or Syt-1 spots was markedly (2–4 fold) higher than the number of CgA-positive vesicles (*Figure 10B,F*). The diameter of CgA-spots (Materials and methods) was estimated to be 171 and 176 nm, respectively, for the two data sets in *Figure 10A–D* and *Figure 10E–H*, which is consistent with the size of vesicles as detected by high-resolution 3D-ET (175.5 nm, *Figure 8R*). Thus, CgA-spots correspond to LDCVs, whereas Syt-1 and Syt-7 spots are clearly distinct.

When inspecting the images, we realized that Syt-1 and Syt-7 spots were found closely associated to CgA-vesicles (*Figure 10A,E*, ROIs 1 and 2). We therefore estimated the distance (center to center) in 3D from CgA-positive structures to the nearest neighboring object in the other channel, either Syt-1 or Syt-7, using the DiAna-plugin for ImageJ (*Gilles et al., 2017*). The distribution of nearest-neighbor distances took the shape of a Gaussian, for both Syt-1 and Syt-7 (*Figure 10D,H*). Since this is the center-to-center distance, it shows that Syt-immunoreactivity is excluded from the center of the vesicle, but found just outside of the vesicle itself, consistent with the fact that the Syt-antibodies detect the cytoplasmic domains of the Syts. We can reach a rough estimate for the minimum distance we expect between a CgA-positive vesicle and a Syt-spot localized immediately adjacent to it (but not overlapping), by adding the estimated mean radii of the CgA and Syt-positive structures (note that the Syt-linker and the size of the antibodies might increase the distance). We find that the minimal distance is 147 nm for CgA vs Syt-7 and 145 nm for CgA vs Syt-1 (indicated on *Figure 10D, H*). Thus, strikingly, the peak of the nearest neighbor distances is found just beyond this distance, and >90% of the vesicles has a Syt-1 or Syt-7 spot within a total distance of 320 nm (from center to center; note the cumulative curves in *Figure 10D,H*). To investigate the consequences of partially randomizing the spots, we repeated the analysis while inverting the CgA-stack top-to-bottom. This resulted in markedly larger nearest-neighbor distances between CgA and Syt-spots (*Figure 10D,H*, orange bars), indicating that CgA and Syt-spots are placed closer together than expected by chance.

Thus, Syt-1 and Syt-7 display poor colocalization as they largely localize to different clusters, but they are both associated with dense-core vesicles, consistent with the localization of both Syt-1 and Syt-7 to a majority of dense-core vesicles. This does not rule out that some vesicles could harbor only or predominantly one of the two Syt isoforms.

## Discussion

### Functional interactions between synaptotagmins

By expressing either Syt-1 or Syt-7 in Syt-1/Syt-7 DKO cells, we showed that both synaptotagmins are able to act as stand-alone calcium sensors, with Syt-7 being a slower sensor than Syt-1 both on the population level, and at the level of single LDCV fusion (*Figure 1*). Kinetically, the two sensors largely give rise to two vesicle pools, the SRP (synaptotagmin-7) and the RRP (synaptotagmin-1). Thus, a simple interpretation is that the RRP and the SRP – being equipped with separate Ca$^{2+}$-sensors – fuse independently. However, further work made it clear that this picture is insufficient, as the two sensors interact. In flash photolysis experiments, Syt-7 KO cells displayed faster fusion kinetics for both RRP and SRP fusion (*Figure 2H*) than the Syt-7 WT, and this effect was rescued by Syt-7

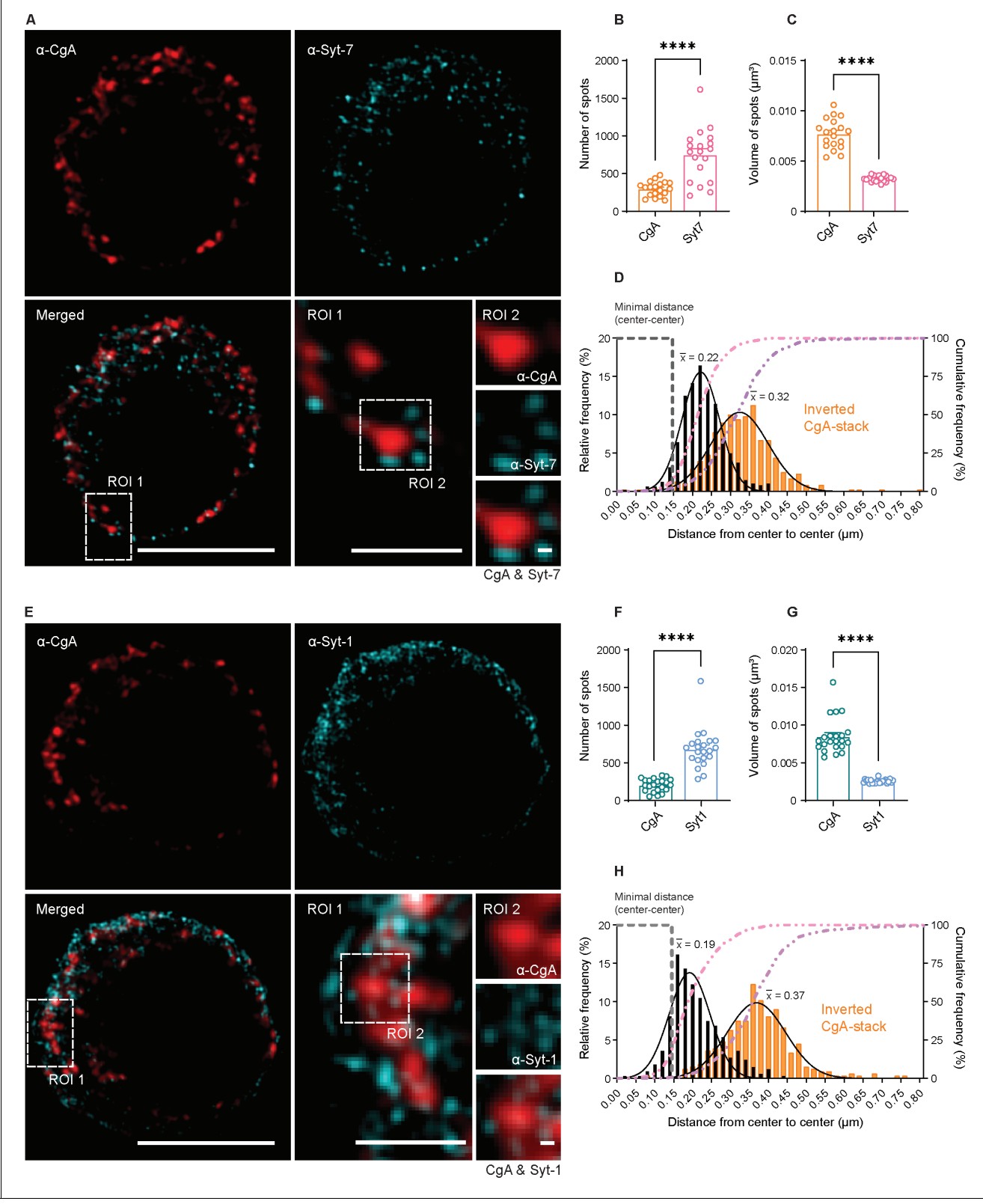

**Figure 10.** Syt-1 and Syt-7 are found on the outside of Chromogranin-A-positive vesicles. (A, E) Single optical slices of WT mouse chromaffin cells stained against CgA (α-CgA) and Syt-7 (α-Syt-7) or Syt-1 (α-Syt-1) acquired with 3D-structured illumination microscopy (3D-SIM). Scale bar: 5 μm. Bottom right: Magnified ROIs. ROI 1: a region of the cell from the merged-channel image. Scale bar: 2 μm. ROI 2: An example of a CgA-vesicles with Syt-7 or Syt-1 spots localized adjacent to it (top panels: α-CgA; middle panels: α-Syt-7 or α-Syt-1; bottom panels: merged image of CgA and Syt-7 or

*Figure 10 continued on next page*

*Figure 10 continued*

Syt-1 channels). Scale bar: 0.2 µm. (B, F) Quantification of spot numbers (mean ± SEM) for Syt-1, Syt-7 and CgA in WT and Syt-7 KO cells using the DiAna plugin for ImageJ. The number of spots was quantified in a 3D-volume with z-length of 2.1 µm around the middle of the cell (note that the numbers in *Figure 9* are for the entire cell and estimated using another plugin). (C, G) Volumes of Syt-1, Syt-7 and CgA spots (mean ± SEM) calculated using the DiAna plugin. (D, H) Center-to-center distances between segmented spots from the CgA-channel (CgA) and their closest neighbor in the Syt-7 or Syt-1 channel. The histogram of distance distribution (black) shows that the center of the nearest Syt-7 and Syt-1 spot is localized at 0.22 and 0.19 µm from the center of CgA vesicles, respectively (mean of Gaussian fits). The histogram of partially randomized distance distributions (orange) obtained after inversion of the CgA-stack shows that the center of the nearest Syt-7 and Syt-1 spot are localized at a larger distance (0.32 and 0.37 µm, respectively, mean of Gaussian fits). A gray dashed line identifies the minimal distance center-to-center between two spots localized side-by-side, calculated by adding the mean radii of CgA- and Syt-spots. A pink dashed line represents the cumulative distribution of the center-to-center distances between spots from the two images. A purple dashed line represents the cumulative distribution of the randomized distances between spots. Data information: Data are presented as mean ± SEM. ****$p < 0.0001$ (Mann-Whitney test). Number of cells: CgA/Syt-7: N = 19 cells; CgA/Syt-1: N = 22 cells. The online version of this article includes the following figure supplement(s) for figure 10:

**Figure supplement 1.** Proposed role of Syt-7 and ubMunc13-2 in dense-core vesicle priming.

overexpression, indicating that the slow kinetics of Syt-7-driven fusion affects the fusion kinetics of both pools. In addition, Syt-7 stimulates the RRP size, by enabling $Ca^{2+}$-dependent priming (*Figure 3*) and by assisting ubMunc13-2 (*Figure 7*) and phorbolester-dependent (*Figure 6*) priming. Note that $Ca^{2+}$-dependent priming requires 10 s of seconds (*Voets, 2000*), and phorbolester 100 s of seconds to develop (*Smith et al., 1998*); thus, the effect of Syt-7 on fast fusion is acute and not likely to be caused by indirect effects on vesicle biogenesis.

RRP fusion requires Syt-1 (*Nagy et al., 2006*; *Voets et al., 2001*) - see also *Figure 1*. Thus, the two sensors act competitively (in setting fusion speeds) and cooperatively (during vesicle priming, where Syt-7 stimulates the number of vesicles fusing via Syt-1), which is hard to reconcile with the simple case of two independent vesicle pools. This agrees with earlier work on chromaffin cells showing that refilling of the RRP happens at the expense of the SRP (*Voets et al., 1999*), and that the effect of mutation in SNAREs is best explained by both slow and fast fusion along a single pathway (*Walter et al., 2013*). Overexpression of Syt-1 or Syt-7 could cause 'overflow' and targeting to other cellular compartments; therefore it is important that in our work the findings of competition and cooperation could be made relying on endogenously expressed Syt-1 and Syt-7, by comparing Syt-7 WT and KO cells. Nevertheless, overexpression of Syt-7 rescued Syt-7 KO secretion amplitude and kinetics, and $Ca^{2+}$-dependent priming (*Figures 2–3*).

Isoform-specific stainings performed by others (*Bendahmane et al., 2020*) and us, indicate that the overlap between endogenous Syt-1- and Syt-7-positive structures (spots) is very limited, a puzzling fact given the functional interaction between the two $Ca^{2+}$-sensors. We performed 3D-SIM, which showed that CgA-containing vesicles are in fact distinct from both Syt-1 and Syt-7 spots; notably, the CgA-vesicles are markedly larger than the Syt spots, whereas the Syt-spots are more numerous. Distance calculations showed that Syt-1 and Syt-7 spots (detected by antibodies recognizing the cytoplasmic Syt-domains) accumulate outside the CgA-vesicle. The distribution of distances is similar for Syt-1 and Syt-7, and increased upon randomization (inversion of the CgA-stack), consistent with the notion that most CgA-vesicles harbor both Syt-1 and Syt-7. The distance is consistent with the Syts being anchored in the vesicle, although other possibilities remain; for instance Syts might be present in small transport vesicles, to be delivered to the CgA-vesicle shortly before fusion (*Hummer et al., 2020*; *Dembla and Becherer, 2021*; *Walter et al., 2014*).

Functional interaction between a fast synaptotagmin (Syt-1, 2, or 9 [*Xu et al., 2007*]) and Syt-7 is also described in neurons, where Syt-7 is a prerequisite for synaptic facilitation (*Jackman et al., 2016*; *Turecek et al., 2017*), which involves stimulation of fast release by Syt-7 – note different findings in *Drosophila* neuromuscular junction (*Guan et al., 2020*). Also the finding that Syt-7 stimulates vesicle replenishment (*Liu et al., 2014*) implies functional interaction. However, in neurons Syt-7 is described as a plasma membrane sensor (*Sugita et al., 2001*), and imaging of pHluorin-tagged Syt-7 showed little, or delayed, exocytosis (*Dean et al., 2012*; *Li et al., 2017*; *Weber et al., 2014*). At higher stimulation strengths, plasma membrane localized Syt-7-pHluorin was initially endocytosed, and then recycled back to the plasma membrane (*Li et al., 2017*). Thus, the localization of Syt-7 in neurons differs from chromaffin cells; this difference might account for the finding (*Figure 2—figure supplement 2* and *Schonn et al., 2008*) that the $Ca^{2+}$-binding sites in the C2B-domain are critical

for Syt-7 function in chromaffin cells, while being dispensable for neuronal function (*Bacaj et al., 2013*).

## Syt-7 in Ca$^{2+}$-dependent priming of fast and slow release

The presence of Syt-7 at endogenous levels caused a larger RRP and SRP size (here and *Schonn et al., 2008*) – and twofold Syt-7 overexpression increased RRP [and SRP] size even more [*Figure 2G*]. Titration of the burst (RRP+SRP) as a function of prestimulation [Ca$^{2+}$] revealed strong Ca$^{2+}$-dependent priming in the presence of Syt-7, but not in its absence (*Figure 3*).

A function for Syt-7 in speeding up RRP replenishment in neurons was reported by *Liu et al., 2014*, whereas *Bacaj et al., 2015* found that Syt-7 helps Syt-1 maintaining the size of the RRP. Intuitively one would expect the first result to be due to a difference in priming rate, whereas the latter would seem to imply an effect on the depriming rate. However, replenishment experiments are hard to interpret at synapses, because release can be limited either by the availability of vesicles or the availability of release sites, and in fact both priming and depriming rates affect pool sizes as well as recovery kinetics (Material and methods, *Equation 13*). Taking advantage of chromaffin cells (where priming sites are not limiting), we could show that Syt-7 has two effects: it decreases the depriming rate and it increases the priming rate following stimulation (*Figure 4A–C*; *Table 1*). In both cases, the effect is approximately a factor of two. The physiological consequence is an upregulation of the priming rate following stimulation to compensate for partial depletion of upstream vesicle pools. In chromaffin cells, the SRP and the RRP are arranged sequentially, with vesicles passing through the SRP to reach the RRP state (*Voets, 2000*; *Walter et al., 2013*). The recovery of the SRP+ RRP (600 ms data point, *Figure 4*) therefore probes the upstream priming step (the 'priming I' step filling the SRP [*Houy et al., 2017*]). The incomplete recovery of the RRP (60 ms points, *Figure 4*) in the Syt-7 KO, and the increased size of the RRP upon Syt-7 overexpression might therefore be secondary to changes in SRP recovery and pool size.

Recalculating the consequences of eliminating Syt-7 in a release site model (*Figure 4D–E*), we showed that our findings are fully consistent with neuronal data showing overall slower recovery in Syt-7 KO neurons (*Chen et al., 2017*; *Liu et al., 2014*). The advantage of chromaffin cells is that we can derive priming and depriming rates separately, demonstrating that they are both affected by Syt-7.

## Syt-7 places dense-core vesicles at the plasma membrane

High-pressure-freezing 3D electron tomography in combination with classical 2D-EM showed that the total number of vesicles as well as the number of vesicles localized within 40 nm of the plasma membrane (i.e. membrane-proximal vesicles) were normal in the Syt-7 KO. Moreover, the number of membrane-attached vesicles (within 0–4 nm of the plasma membrane) was lower, although not significantly altered in Syt-7 KO chromaffin cells (*Figure 8P,T*). In line with our previous findings for the analysis of Munc13-deficient chromaffin cells (*Man et al., 2015*), the calculated number of docked LDCVs per cell in the present study (~329 in WT cells) exceeds the number of vesicles in the functional RRP (~50 vesicles, according to *Figure 2A–B* with 0.94 fF/vesicle [*Pinheiro et al., 2014*]) and SRP (~30 vesicles) at resting Ca$^{2+}$-concentrations, thereby supporting the notion that functionally primed LDCVs in chromaffin cells are difficult to identify by morphological analysis due to a large number of dead-end docked LDCVs that are morphologically docked, but incapable of fusion upon stimulation (*Hugo et al., 2013*; *Verhage and Sørensen, 2008*). Indeed, it has been shown that by preventing full SNARE-zippering, synaptic vesicles can be rendered in a docked, but fusion-incompetent state (*Gipson et al., 2017*; *Vardar et al., 2016*). Interestingly, we found a significant reduction in the number of LDCVs within 0–6 nm of the plasma membrane in Syt-7 KO chromaffin cells. In vitro studies have shown that Munc13-1 forms a ~ 20 nm elongated structure that can bridge membranes (*Quade et al., 2019*; *Xu et al., 2017*) and individual SNARE-proteins can interact at distances smaller than 8 nm (*Gao et al., 2012*) or within 15 nm in the presence of SNARE-regulators like complexins (*Li et al., 2011*). The reduction of LDCVs within 6 nm of the plasma membrane in Syt-7 KO cells, which was accompanied by an increase in vesicle numbers within 20–40 nm, indicates a role for Syt-7 in mediating or stabilizing distance-dependent SNARE/Munc13-mediated interactions, thereby regulating vesicle priming and depriming. The increased number of more distant vesicles could

account for the finding of a higher forward priming rate into the SRP in Syt-7 KO (*Table 1*, the 600 ms data point; see also *Durán et al., 2018*).

A similar function for Syt-1 as a $Ca^{2+}$-dependent distance-regulator between fusing membranes was previously proposed (*Chang et al., 2018*; *van den Bogaart et al., 2011*). In chromaffin cells, EM data obtained using chemical fixation methods implicated Syt-1 in vesicle docking (*de Wit et al., 2009*; *Kedar et al., 2015*). Although data obtained using chemical fixation cannot be compared directly to HPF-data here and (*Man et al., 2015*), overall these data indicate that either Syt-1 or Syt-7 might adopt the role of placing LDCVs into the vicinity of the plasma membrane to promote or stabilize SNARE/Munc13-interactions and thereby vesicle priming.

## Interdependence of Syt-7 and ubMunc13-2/phorbolester-dependent priming

We investigated how the priming role of Syt-7 interacts with ubMunc13-2, the dominating Munc13 protein in chromaffin cells (*Man et al., 2015*), which acts in the same priming step as Syt-7, upstream of the SRP. Treatment of Syt-7 KO chromaffin cells with phorbolesters (which activate Munc13 proteins via their C1-domain) at low prestimulation $[Ca^{2+}]$ caused only a modest increase in RRP and SRP (overall release was potentiated by 21%), whereas phorbolesters caused a 126% increase in WT cells (*Figure 6*). Likewise, in ubMunc13-2 overexpressing cells, the RRP was smaller in Syt-7 KO cells compared to WT cells (*Figure 7*). These data show that the priming roles of Syt-7 and ubMunc13-2 are interdependent. To explain these data, as well as the EM-findings, we propose that a population of vesicles is attached to the plasma membrane via Syt-7. This brings the vesicles within the critical distance (6–10 nm) of the plasma membrane, where Munc13-proteins are able to bridge the gap between vesicle and plasma membrane, which allows SNARE-complex formation and priming (*Figure 10—figure supplement 1*). In the Syt-7 KO, these vesicles reside at longer distances (20–40 nm), which prevents the formation of the SNARE-complex even when ubMunc13-2 is overexpressed. Interestingly, ubMunc13-2 overexpression stimulated a large SRP, but this SRP fused after a delay in almost all Syt-7 KO cells, whereas in Syt-7 WT cells - especially at higher prestimulation $[Ca^{2+}]$ - the SRP fused without delay. Thus, Syt-7 probably causes a movement toward the membrane, which is reinforced by binding to $Ca^{2+}$; when Syt-7 is absent, vesicles reside at longer distances and fusion is therefore delayed. This can also explain the limited effect of treating Syt-7 KO cells with phorbolesters (*Figure 6*), which anchor Munc13-2 into the plasma membrane. This will be ineffective if vesicles are out of reach of the Munc13-2/SNARE-complex. Finally, N-ethylmaleimide caused an increase in RRP size in Syt-7 KO cells when stimulated from a low prestimulation $[Ca^{2+}]$ (*Figure 5*). This is consistent with our refilling experiments (*Figure 4*) and indicates that a function of Syt-7 is to protect against depriming, either by binding directly to SNARE-proteins (*Bacaj et al., 2015*), or by moving vesicles closer to the membrane, where they can interact with Munc13-2, which in turn protects against depriming (*He et al., 2017*; *Ma et al., 2013*; *Prinslow et al., 2019*). The effect of NEM was only seen at low prestimulation $[Ca^{2+}]$, which indicates that other mechanisms stabilize the RRP at higher calcium concentrations. Overall, all three experiments (NEM, PMA, and ubMunc13-2) are consistent with the misalignment of vesicles with the plasma membrane observed in the Syt-7 KO.

Although $Ca^{2+}$-dependent priming per se is almost absent in the Syt-7 KO, the combination of Syt-7 KO with high prestimulation $[Ca^{2+}]$ and other manipulations to increase Munc13-priming (ubMunc13-2, PMA) allow priming in the Syt-7 KO, which is consistent with the presence of multiple priming pathways in chromaffin cells (*Liu et al., 2010*). If the main function of Syt-7 is to deliver vesicles to the membrane (and to Munc13) other calcium sensors that carry out similar functions might compensate for the absence of Syt-7. In chromaffin cells, Syt-1 (*Nagy et al., 2006*; *Voets et al., 2001*), and Doc2B (*Houy et al., 2017*; *Pinheiro et al., 2013*) are the best candidates. As mentioned above, Syt-1 has been implicated in a similar priming role, also in neurons (*Chang et al., 2018*; *Ruiter et al., 2019*). Expression of membrane-bound Doc2B causes priming to saturate at the maximal level in chromaffin cells (*Friedrich et al., 2008*; *Houy et al., 2017*), that is the $Ca^{2+}$-dependent priming process, which normally depends on Syt-7, is overwhelmed, rendering priming $Ca^{2+}$-independent, although $Ca^{2+}$-dependent priming is still present in the Doc2B KO (*Pinheiro et al., 2013*).

We conclude that Syt-7 mediates Munc13- and $Ca^{2+}$-dependent vesicle priming and vesicle fusion in a competitive and cooperative interplay with Syt-1, and that these functions involve positioning of vesicles in close membrane apposition.

# Materials and methods

## Key resources table

| Reagent type (species) or resource | Designation | Source or reference | Identifiers | Additional information |
|---|---|---|---|---|
| Strain, strain background (*M. musculus*) | C57BL/6 | Experimental Medicine, Panum Stable, University of Copenhagen. | | |
| Strain, strain background (*M. musculus*) | CD1 | Experimental Medicine, Panum Stable, University of Copenhagen. | | |
| Genetic reagent (*M. musculus*) | Synaptotagmin-7 (syt7) null allele | Maximov A, Lao Y, Li H, Chen X, Rizo J, Sørensen JB, Südhof TC. Genetic analysis of synaptotagmins-7 function in synaptic vesicle exocytosis. Proc Natl Acad Sci U S A. 2008 Mar 11;105(10):3986–3991. | PMID:18308933 | |
| Genetic reagent (*M. musculus*) | Synaptotagmin-1 (syt1) null allele | Geppert M, Goda Y, Hammer RE, LI C, Rosahl TW, Stevens CF, Südhof TC. 1994. Synaptotagmins I: a major Ca2+ sensor for transmitter release at a central synapse. Cell 79(4): 717–727. | PMID:7954835 | |
| Transfected construct (*Rattus norwegicus*) | p156rrl-pCMV- pH (ecliptic GFP)-TEV-rnSyt1 | This paper, Syt-1 WT | | Local reference: Lenti #94 |
| Transfected construct (*Rattus norwegicus*) | p156rrl-pCMV- pH (ecliptic GFP)-TEV-rnSyt7S | This paper, Syt-7 WT | | Local reference: Lenti #96 |
| Transfected construct (*Rattus norwegicus*) | p156rrl-pCMV- pH (ecliptic GFP)-TEV-rnSyt7S-C2AB/D225,227,233,357,359A | This paper, Syt-7 C2AB* | | Local reference: Lenti #133 |
| Transfected construct (*Rattus norwegicus*) | p156rrl-pCMV- pH(ecliptic GFP)-TEV-rnSyt7S-C2B/D357,359A | This paper, Syt-7 C2B* | | Local reference: Lenti #135 |
| Transfected construct (*Rattus norwegicus*) | p156rrl-pCMV- pH (ecliptic GFP)-TEV-rnSyt7S-C2A/D225, 227,233A | This paper, Syt-7 C2A* | | Local reference: Lenti #136 |
| Transfected construct (*Rattus norwegicus*) | pSFV1-EGFP-ubMunc13-2 | Zikich D, Mezer A, Varoqueaux F, Sheinin A, Junge HJ, Nachiel E, Melamed R, Brose N, Gutman M, Ashery U. 2008. J. Neurosci. 28:1949–1960. | PMID:18287511 | Local reference: Semliki #486 |
| Antibody | Chicken anti-GFP | Abcam | Ab13970 RRID:AB_300798 | 1:500; 2 hr at room temperature |
| Antibody | Rabbit anti-Chromogranin A | Abcam | Ab15160 RRID:AB_301704 | 1:500; Overnight at four degrees |
| Antibody | Rabbit anti-synaptotagmin-1 | Gift from T. C. Südhof, Stanford, CA | W855 | 1:2000; 2 hr at room temperature |
| Antibody | Mouse anti-synaptotagmin-1 | Synaptic System | SySy: 105011 RRID:AB_887832 | 1:500; Overnight at 4 degrees or 2 hr at room temperature |
| Antibody | Rabbit anti-synaptotagmin-7 | Synaptic System | SySy: 105173 RRID:AB_887838 | ICC: 1:500; Overnight at 4 degrees or 2 hr at room temperature WB: 1:500; Overnight at four degrees. |
| Antibody | Mouse anti-synaptotagmin-7 | Sigma-aldrich | MABN665 RRID:AB_2888943 | 1:200; Overnight/4 degrees |

*Continued on next page*

*Continued*

| Reagent type (species) or resource | Designation | Source or reference | Identifiers | Additional information |
|---|---|---|---|---|
| Antibody | anti-VCP | Abcam | Ab11433 RRID:AB_298039 | 1:10000; 1 hr at room temperature |
| Antibody | Goat anti-rabbit HRP | Agilent | Dako-P0448 RRID:AB_2617138 | 1:2000; 1 hr and 30 min at room temperature |
| Antibody | Goat anti-mouse HRP | Agilent | Dako-P0447 RRID:AB_2617137 | 1:10000; 30 mins at room temperature |
| Antibody | Goat anti-mouse Alexa 546 | ThermoFisher Scientific | A11003 AB_2534071 | 1:500; 30 min at room temperature |
| Antibody | Goat anti-rabbit Alexa 647 | ThermoFisher Scientific | A21245 RRID:AB_2535813 | 1:500; 30 min at room temperature |
| Antibody | Goat anti-mouse Alexa 488 | Invitrogen | A11029 RRID:AB_2534088 | 1:500; 30 min at room temperature |
| Antibody | Goat anti-chicken Alexa 488 | Abcam | Ab150169 RRID:AB_2636803 | 1:500; 30 min at room temperature |
| Commercial assay or kit | BCA Protein assay kit | Pierce | Pierce: 23227 | |
| Chemical compound, drug | NaCl | Sigma-aldrich | Sigma-aldrich: S9888 | |
| Chemical compound, drug | KCl | Sigma-aldrich | Sigma-aldrich: P5405 | |
| Chemical compound, drug | $NaH_2PO_4$ | Sigma-aldrich | Sigma-aldrich: S8282 | |
| Chemical compound, drug | Glucose | Sigma-aldrich | Sigma-aldrich: G8270 | |
| Chemical compound, drug | DMEM | Gibco | Gibco: 31966047 | |
| Chemical compound, drug | L-cysteine | Sigma-aldrich | Sigma-aldrich: C7352 | |
| Chemical compound, drug | $CaCl_2$ | Sigma-aldrich | Sigma-aldrich: 499609 | |
| Chemical compound, drug | EDTA | Sigma-aldrich | Sigma-aldrich: E5134 | |
| Chemical compound, drug | Papain | Worthington Biochemical | Worthington Biochemical: LS003126 | |
| Chemical compound, drug | Albumin | Sigma-aldrich | Sigma-aldrich: A3095 | |
| Chemical compound, drug | Trypsin-inhibitor | Sigma-aldrich | Sigma-aldrich: T9253 | |
| Chemical compound, drug | Penicillin/streptomycin | Invitrogen | Invitrogen: 15140122 | |
| Chemical compound, drug | Insulin-transferrin-selenium-X | Invitrogen | Invitrogen: 51500056 | |
| Chemical compound, drug | Fetal calf serum | Invitrogen | Invitrogen: 10500064 | |
| Chemical compound, drug | $MgCl_2$ | Sigma-aldrich | Sigma-aldrich: 449172 | |
| Chemical compound, drug | HEPES | Sigma-aldrich | Sigma-aldrich: H3375 | |
| Chemical compound, drug | Nitrophenyl-EGTA (NPE) | Synthesized at the Max-Planck-Institut for biophysical chemistry, Göttingen. | | |

*Continued on next page*

*Continued*

| Reagent type (species) or resource | Designation | Source or reference | Identifiers | Additional information |
|---|---|---|---|---|
| Chemical compound, drug | Fura-4F | Invitrogen | Invitrogen: F14174 | |
| Chemical compound, drug | Furaptra | Invitrogen | Invitrogen: M1290 | |
| Chemical compound, drug | Mg-ATP | Sigma-aldrich | Sigma-aldrich: A9187 | |
| Chemical compound, drug | GTP | Sigma-aldrich | Sigma-aldrich: G8877 | |
| Chemical compound, drug | Ascorbic acid | Sigma-aldrich | Sigma-aldrich: A5960 | |
| Chemical compound, drug | EGTA | Sigma-aldrich | Sigma-aldrich: E4378 | |
| Chemical compound, drug | Ethylmaleimide (NEM) | Sigma-aldrich | Sigma-aldrich: 04259 | |
| Chemical compound, drug | Phorbol 12-myristate 13-acetate (PMA) | Sigma-aldrich | Sigma-aldrich: P8139 | |
| Chemical compound, drug | Paraformaldehyde | Sigma-aldrich | Sigma-aldrich: P6148 | |
| Chemical compound, drug | PIPES | Sigma-aldrich | Sigma-aldrich: 80635 | |
| Chemical compound, drug | Triton X-100 | Sigma-aldrich | Sigma-aldrich: T8787 | |
| Chemical compound, drug | BSA | Sigma-aldrich | Sigma-aldrich: A4503 | |
| Chemical compound, drug | Prolong Gold | Invitrogen | Invitrogen: P36934 | |
| Chemical compound, drug | Protease cocktail inhibitor | Invitrogen | Invitrogen: 87785 | |
| Chemical compound, drug | RIPA buffer | Invitrogen | Invitrogen: R0278 | |
| Chemical compound, drug | ECL plus western blotting substrate | Pierce | Pierce: 32132 | |
| Software, algorithm | Igor 8.0 | Wavemetrics | | |
| Software, algorithm | ImageJ | NIH software | | |

## Mouse lines and cell culture

Mouse lines (C57/Bl6-Syt-1, *Geppert et al., 1994*, C57/Bl6-Syt-7 [*Maximov et al., 2008*]) were kept in an AAALAC-accredited stable at the University of Copenhagen operating a 12 hr/12 hr light/dark cycle with access to water and food ad libitum. Permission to keep and breed KO mice were obtained from the Danish Animal Experiments Inspectorate (permissions 2006/562–43 and 2018-15-0202-00157). Primary chromaffin cell cultures were prepared as described (*Sørensen et al., 2003*). Syt-7 knockout (KO) cells were obtained from P0-P2 pups of either sex originating from Syt-7 heterozygous crossing and identified by PCR genotyping (*Maximov et al., 2008*). Syt-1/Syt-7 DKO cells were obtained from embryos of either sex at embryonic day 18 (E18) by crossing mice that were homozygous KO (-/-) for Syt-7 and heterozygous (+/-) for Syt-1 (*Schonn et al., 2008*). The embryos were PCR genotyped for both Syt-1 and Syt-7. Cells marked Wild Type (WT) were P0-P2 WT littermates of the Syt-7 line, unless otherwise noted. The adrenal glands were dissected and cleaned in Locke's solution consisting of (in mM) 154 NaCl, 5.6 KCl, 0.85 $NaH_2PO_4$, 2.15 $Na_2HPO_4$, and 10 glucose, and adjusted to pH 7.0. The glands were digested with 20–25 units/ml of papain enzyme for 45 min at 37°C and 8% $CO_2$ followed by 10–15 min inactivation with DMEM-Inactivation solution. To

dissociate the cells, the glands were gently triturated and 40–50 µl of the cell suspension was placed onto the center of the coverslip, incubated to settle for 30 min and finally supplemented with 1–2 ml enriched DMEM media. Cells were used 2–5 days after plating. DMEM-Papain solution contained (in mM) (1 CaCl$_2$, 0.5 EDTA) supplemented with 0.2 mg/ml l-cysteine and equilibrated with 8% CO$_2$. DMEM-Inactivation solution contained 10% heat-inactivated FCS, 2.5 mg/ml albumin, and 2.5 mg/ml trypsin inhibitor and equilibrated with 8% CO$_2$. DMEM culture medium consisted of DMEM supplemented with 4 µl/ml penicillin/streptomycin and 10 µl/ml insulin-transferrin-selenium-X and equilibrated with 8% CO$_2$.

## Viral constructs

The experiments made use of N-terminal pHluorin-tagged Syt-1 and Syt-7 (the α isoform [*Fukuda et al., 2002*]) constructs. The pHluorin (ecliptic EGFP) was preceded by a signal sequence of preprotachykinin to ensure correct orientation of the fusion protein into the vesicle membrane (*Diril et al., 2006*). The pHluorin-rat(rn)Syt cassette was cloned into the multiple cloning site of a lentiviral vector containing CMV promotor and a downstream WPRE sequence. To eliminate Syt-7 calcium binding, the C2 domains mutated fragments carrying aspartate/alanine exchange (C2A*: D225, 227, 233A; C2B*: D357, D359A) (*Bacaj et al., 2013*) were synthesized (Invitrogen GeneArt Gene Synthesis) and subcloned into the Syt-7 WT construct. In total, five constructs were produced: phluorin-rnSyt1; phluorin-rnSyt7α; phluorin-rnSyt7 α/C2A*; phluorin-rnSyt7 α/C2B*; phluorin-rnSyt7 α/C2AB*. All constructs were verified by sequencing. Lentiviruses were produced according to standard protocols using Lipofectamine2000 transfection and a HEK293FT cell host. The pHluorin-tags were used for identifying expressing cells. For lentiviral expression, cells were transduced 24 hr after prepping and incubated for 46–50 hr before being used for experiments. Acute expression, 4–6 hr, of EGFP-fused ubMunc13-2 was induced from a Semliki Forest Virus construct (*Zikich et al., 2008*).

## Immunostaining, confocal and structural illumination microscopy

Cells were plated on 25 mg/ml poly-D-lysine (Sigma P7405) coated coverslips. Prior to fixation, cells were cooled on ice for 3–5 min and fixated with ice cold 4% Paraformaldehyde (PFA; EMC 15710) for 15 min on ice, in the presence or absence of 0.2% Glutaraldehyde (Merck Millipore 104239), followed by 2% PFA for an additional 10 min at room temperature (RT). Cells were permeabilized with 0.15% Triton-X100 (Sigma-Aldrich T8787) for 15 min at RT and subsequently blocked with 0.2% cold fish gelatin (Sigma-Aldrich G7765), 1% goat serum (Thermo Fisher Scientific 16210064) and 3% Bovine Serum Albumin (Sigma-Aldrich A4503) for 1 hr at RT. Cells were washed with PBS and glutaraldehyde autofluorescence was quenched with 0.1% Sodium Borohydride (NaBH4; Sigma-Aldrich 213462). Primary and secondary antibodies were diluted in blocking solution, with different antibody combinations being applied for the different immunostainings.

Stainings against Syt-1 and Syt-7 were performed using either the primary rabbit polyclonal α-Syt7 (1:500; Synaptic Systems 105173) and mouse monoclonal α-Syt1 (1:500; Synaptic Systems 105011) antibodies (*Figure 9A,D*; *Figure 9—figure supplement 1A–C*) or the rabbit polyclonal α-Syt1 (1:2000; W855; a gift from T. C. Südhof, Stanford, CA) and the mouse monoclonal α-Syt7, clone 275/14 (1:200; Sigma-Aldrich MABN655) antibodies (*Figure 9—figure supplement 1D–F*) for 2 hr at RT. Secondary antibodies goat α-mouse Alexa Fluor 488 conjugate (1:500, Thermo Fisher Scientific A11029 and goat α-rabbit Alexa Fluor 647 conjugate (1:500; Thermo Fisher Scientific A21245) were incubated at RT for 30 min in both staining setups (*Figure 9—figure supplement 1D–F*)).

Stainings against Syt-1, Syt-7 and Syt-7 constructs mutated in the C2A and/or C2B (*Figure 2—figure supplement 2*) were performed with chicken polyclonal α-GFP (1:500; Abcam ab13970), rabbit polyclonal α-Syt7 (1:500; Synaptic Systems 105173) and mouse monoclonal α-Syt1 (1:500; Synaptic Systems 105011). A combination of goat α-chicken Alexa Fluor 488 conjugate (1:500; Abcam ab150169), goat α-mouse Alexa Fluor 546 conjugate (1:500, Thermo Fisher Scientific A11003) and goat α-rabbit Alexa Fluor 647 conjugate (1:500; Thermo Fisher Scientific A21245) secondary antibodies was used for 30 mins at RT.

For costaining of chromogranin A with syts (*Figure 10*), no Glutaraldehyde was added during fixation. Rabbit polyclonal α-CgA (1:500; Abcam ab15160) and mouse monoclonal α-Syt7, clone 275/14 (1:200; Sigma-Aldrich MABN655) or mouse monoclonal α-Syt1 (1:500; Synaptic Systems 105011), were incubated overnight at 4°C. Secondary antibodies used were goat α-mouse Alexa Fluor 488

conjugate (1:500, Thermo Fisher Scientific A11029) and goat α-rabbit Alexa Fluor 647 conjugate (1:500; Thermo Fisher Scientific A21245).

## Confocal microscopy

Immunofluorescence was visualized using a Zeiss LSM 780 inverted confocal with oil-immersion Plan-Apochromat NA 1.4 63x objective. The fluorophores were excited with Argon 488 nm (25 mW), HeNe 543 nm (1.2 mW), 633 nm (5 mW), and intune 488–645 nm (5 mW) lasers. Linear unmixing was applied on cells stained with more than two fluorophores in Zen Black Zeiss software. Control cells stained with a single fluorophore were used to define the spectral finger print. Quantification of Syt-1, Syt-7 was performed on ImageJ software on average projections of 0.5 μm increment z-stacks. Mean intensity of circular cell ROI was background subtracted and the value was normalized to control cells (wild-type cells or Syt7-KO cells). Control cells were acquired on the same day as sample cells, and laser power, gain and emission detection were unchanged. The Manders' Coefficient was calculated on single z-stacks, using the JACoP plugin for ImageJ (*Bolte and Cordelières, 2006*).

## 3D-structural illumination microscopy (3D-SIM)

For costaining of Syt-1 and Syt-7 (*Figure 9D–H*) images with voxel size (x,y,z) in μm: 0.03 × 0.03×0.11 were obtained with a Zeiss Elyra PS.1 microscope equipped with a sCMOS PCO.edge camera and an oil-immersion Plan-Apochromat NA 1.4 63x objective. Alexa Fluor −488 and −647 were excited with 488 HR Diode-200mW and HR Diode-150mW lasers, respectively. Distance-based co-localization analysis between Syt-1 and Syt-7 was performed using Spots colocalization (ComDet) ImageJ plugin (available online, authored by Eugene Katrukha). Vesicles were automatically detected in their respective channels (Syt-1 and Syt-7) by setting the detection threshold to 3xSD above background noise and an approximate particle size of 120 nm. Note that the 120 nm particle size setting allows detection of vesicles that are larger than 120 nm in diameter but restrict the detection of very large undefined structures. Finally, vesicles were scored as co-localized if the coordinates from one channel were within 60 nm distance (half of the set particle size) in the other channel. Since the spots are larger than one imaging plane, they would show up in more than one imaging plane, and we therefore analyzed every third plane, equivalent to 330 nm Z-intervals, from the top to the bottom of the cell. To estimate the number of duplicate spots (same spot that is detected in two subsequent planes), we compared the XY coordinates of vesicles in sequentially analyzed planes and counted matching positions as duplicates. This indicated that the number of duplicates were <1%.

For costaining of Syts with CgA (*Figure 10*), images were obtained using 3D-SIM with the same microscope and settings as in the Syt1/Syt-7 costaining experiment. Object-based 3D-Distance analysis between CgA and either Syt-7 or Syt-1 was performed using DiAna plugin for ImageJ (available online [*Gilles et al., 2017*]). We analyzed a volume consisting of 21 adjacent optical slices around the center of the cell, for a total thickness of 2.1 μm. Objects segmentation was performed by iterative thresholding process (parameters: Min volume: three pixels: Max Volume; 2000 pixels; Min Threshold 500–5000; Step value: 100) and followed by 3D-distance measurements from center-to-center on the segmented images. The distribution of center-to-center distances of each CgA object to the nearest Syt object was displayed in the 3D-distance analysis (*Figure 10D,H*). Number and volume of the CgA, Syt-7 and Syt-1 spots were obtained from the DiAna Plugin after object segmentation. We calculated the radius or diameter of spots from their volume, taking into account that objects in 3D-SIM appear elongated along the Z-axis; we thus assumed objects to be ellipsoids with an aspect ratio (ratio between longest, Z, and shortest, XY, axes) as determined by the calibration of the microscope. The calculated radii or diameters refer to the XY-plane. Since the exact shape of each object is not known, these calculations are only approximate. Randomized 3D-distance measurements were performed by inverting the stack top-to-bottom of the CgA channel and re-performing the frequency distribution analysis of center-to-center distances (orange bar diagram in *Figure 10D,H*). Note that this does not entirely randomize the distances, especially considering vesicles placed near the middle of the stack. We preferred this method, because it preserved the nucleus position, whereas flipping one channel left-to-right would often shift the nucleus from one side of the cell to the other.

## Western blot

Adrenal glands and brain extracts were collected from P0-1 Syt-7 WT and KO mice and lysed in RIPA buffer supplemented with Protease Inhibitor Cocktail (Invitrogen, 89900). The supernatants were collected and protein concentrations were estimated by using the BCA Protein Assay Kit (Pierce 23227) and plotting the resulting BSA curve. 25 μg and 15 μg of protein, from adrenal glands and brain extracts, respectively, were resolved by 4–12% SDS-PAGE (Invitrogen, Thermo Fisher Scientific) and wet-transferred onto an Amersham Hybond LFP PVDF membrane (GE Healthcare). The membrane was blotted with rabbit polyclonal α-Syt7 (1:500; Synaptic Systems SY105173) and mouse monoclonal α-VCP (1:10000; Abcam ab11433), as a loading control, followed by HRP-conjugated α-rabbit (1:2000; Agilent Dako-P0448) and HRP-conjugated α-mouse (1:10000; Agilent Dako-P0447) secondary antibodies. The blot was developed by chemiluminescence SuperSignal West Femto and Pierce ECL Plus Western blotting substrate systems (Thermofisher Scientific) and immunoreactive bands were detected using the FluorChemE image acquisition system (ProteinSimple) equipped with a cooled CCD camera.

## Electrophysiology

Exocytosis was monitored by combining membrane capacitance measurements and carbon fiber amperometry combined with $Ca^{2+}$ uncaging (*Houy et al., 2021*). Capacitance measurements were based on the Lindau-Neher technique using Pulse HEKA software with Lock-In extension. A 70 mV peak-to-peak sinusoid (1000 Hz) was applied around a holding potential of $-70$ mV in the whole-cell configuration. The clamp currents were filtered at 3 kHz and recorded at 12 kHz with an EPC9 HEKA amplifier. Secretion was triggered by 1–2 ms UV flash-photolysis of the caged $Ca^{2+}$ compound nitrophenyl-EGTA, infused through the patch pipette. The UV-flash delivered from a flash lamp (Rapp Optoelectronic, JML-C2) was bandpass-filtered around 395 nm, transmitted through a light guide and a dual condenser and focused with a Fluar 40X/N.A. 1.30 oil objective.

The intracellular $Ca^{2+}$ concentration was determined as described in *Nagy et al., 2002*. Two florescent dyes with different affinities toward $Ca^{2+}$, Fura4F (Kd = 1 μM) and furaptra (Kd = 40 μM) were infused via the pipette into the cell. For ratiometric detection, alternating monochromator excitations of 350 nm and 380 nm were generated at 40 Hz and emission was detected via a photodiode, recorded at 3 kHz and filtered at 12 kHz. The 350/380 ratio was pre-calibrated by infusing the cell with known $Ca^{2+}$ concentrations.

Amperometric recordings were performed as previously described (*Bruns, 2004*) using a carbon fiber (5–10 μm diameter) insulated with polyethylene and mounted in glass pipette. The fiber was clamped at 700 mV, currents were filtered at 5 kHz and sampled at 25 kHz by an EPC7 HEKA amplifier. 50 Hz noise was online eliminated by a Humbug noise eliminator device. When amperometry was performed in parallel with capacitance measurements, we removed the cell from the carbon fiber after measurements and applied another UV-flash stimulation, in order to detect the current caused by light-stimulation of the fiber, in the absence of any catecholamines. The 'empty' amperometric trace was then subtracted from the initial recording to eliminate the photoelectric artifact. Because this artifact depended on the location of the fiber within the stimulation field, we recorded an 'empty' amperometric trace for each cell. For single spike analysis, amperometric traces were off-line filtered at 500 Hz using a Gaussian filter, threshold detected at 5 pA and analyzed with a Igor Pro macro (*Mosharov, 2008*). The interspike interval is the median time interval between the spikes (*Mosharov, 2008*).

The pipette solution contained (in mM): 100 Cs-glutamate, 8 NaCl, 4 $CaCl_2$, 32 Cs-HEPES, 2 Mg-ATP, 0.3 GTP, 5 Nitrophenyl-EGTA (NPE), 0.4 fura-4F, 0.4 furaptra, and 1 ascorbic acid. Adjusted to pH 7.2 and osmolarity to ~295 mOsm. The extracellular solution contained (in mM): 145 NaCl, 2.8 KCl, 2 $CaCl_2$, 1 $MgCl_2$, 10 HEPES, and 11 glucose. Adjusted to pH 7.2 and osmolarity to ~305 mOsm. In some cases, small amounts of $CaCl_2$ or NPE were added to the pipette solution, to adjust the $[Ca^{2+}]$(*Houy et al., 2021*). For the double flash (recovery) experiment (*Figure 4* and *Figure 4—figure supplement 2*), the NPE concentration was reduced to 3 mM, and the $CaCl_2$ was also reduced (to 2.2 mM) to reach a $[Ca^{2+}]$ of 250–350 nM. UV flash intensity was adjusted before the second flash to ensure comparable post-flash calcium levels and fast $Ca^{2+}$ relaxation after the flash. *N*-Ethylmaleimide (NEM) was prepared fresh prior to each experiment and added in the pipette solution to a final concentration of 200 μM. NEM was infused into the cells for 60–100 s before

stimulation. Control cells were treated equally but patched with a pipette solution that did not contain NEM. Phorbol 12-myristate 13-acetate (PMA) was dissolved in DMSO and diluted in extracellular solution immediately prior to the experiment, to a final concentration of 100 nM and used within 2 hr.

## Kinetics analysis

Pool sizes were determined either 0.5 s after the flash and designated as 'Burst' or by fitting the capacitance trace with a sum of two exponentials plus a straight line using a custom written Igor Pro macro (Wavemetrics):

$$\Delta C_m = A_1\left(1 - e^{-t/\tau_1}\right) + A_2\left(1 - e^{-t/\tau_2}\right) + A_3 t \tag{1}$$

where the amplitudes $A_1$ and $A_2$ are the sizes of the releasable pools, and $\tau_1$ and $\tau_2$ are their fusion time constants. The resulting exponential components were assigned as RRP or SRP according to the estimated time constant ($\tau$). Except if noted otherwise, kinetic components were considered to originate from the RRP when $\tau \leq 60$ ms and from the SRP when $60$ ms $\leq \tau \leq 1000$ ms. The sustained release rates were calculated as the linear rate ($A_3$) following fusion of SRP and RRP. In cases where the fit identified two time constants within the same cut off criterion (i.e. both time constants would correspond to either a RRP or the SRP), the trace was refitted with a single exponential for the corresponding component.

In one case (Munc13-2 overexpression in Syt-7 KO and sometimes in Syt-7 WT), capacitance traces had an S-formed shape, which made it impossible to fit them with (*Equation1*). Instead, we here derive a new function for the SRP, which takes into account the delayed fusion of this pool of vesicles. We first observe (in agreement with *Equation 1*) that the fusion of the SRP follows the evolution:

$$\Delta C_m = SRP\left(1 - e^{-t/\tau_{slow}}\right) = SRP\left(1 - e^{-k_{slow} \cdot t}\right) \tag{2}$$

where $\tau_{slow}$ is the time constant of fusion and $k_{slow}$ is the rate constant of fusion and $\tau_{slow} = \frac{1}{k_{slow}}$. Let us assume that the fusion rate of SRP is not instantaneously at its max value, $k_{slow}$; instead, the fusion rate of the SRP, $k_2$, increases gradually toward $k_{slow}$:

$$k_2 = k_{slow}\left(1 - e^{-t/\tau_{delay}}\right) \tag{3}$$

where $\tau_{delay}$ is the time constant of the development of $k_2$ toward the value $k_{slow}$. Therefore, the new model for fusion of the SRP is:

$$\Delta C_m = SRP\left(1 - e^{-t \cdot k_{slow} \cdot \left(1 - e^{-t/\tau_{delay}}\right)}\right) \tag{4}$$

Fitting this equation to capacitance traces is an illformed problem, because $k_{slow}$ and $k_{delay}$ can not both be determined. To see why, we consider the Taylor expansion of the expression for $k_2$:

$$k_2 \approx k_{slow} k_{delay} \cdot t - k_{slow}\left(k_{delay}\right)^2 t^2 + k_{slow}\left(k_{delay}\right)^3 t^3 \ldots \tag{5}$$

If the delay is long (i.e. $k_{delay}$ is small), we can replace $k_2$ with its first-order approximation, $k_{slow} k_{delay} \cdot t$ to yield

$$\Delta C_m = SRP\left(1 - e^{-k_{slow} \cdot k_{delay} \cdot t^2}\right) \tag{6}$$

Or, if we set $\tau_{slow,\,delay} = \frac{1}{k_{slow} \cdot k_{delay}} = \tau_{slow} \cdot \tau_{delay}$ we get

$$\Delta C_m = SRP\left(1 - e^{-t^2/\tau_{slow,\,delay}}\right) \tag{7}$$

Here, $\tau_{slow,\,delay}$ has a different meaning than $\tau_{slow}$, because it includes both the delay and the fusion kinetics of the SRP. For fitting with a delayed SRP, *Equation 7* was substituted into *Equation 1*.

## Priming models

### Model I – no maximal primed vesicle pool size

To understand the consequences of changes to priming and depriming rates, we considered a simple 2-pool model, similar to *Heinemann et al., 1993*, where a large reserve pool ('DP' for Depot Pool) is filling up a primed vesicle pool, ('PP' for Primed Pool), through a reversible priming step, which drives priming forward with rate constant $k_1$, and supports depriming (i.e. the reverse priming reaction) with rate constant $k_{-1}$. Fusion is supported with rate constant $k_f$. Under such conditions, the change in the primed pool is given by:

$$\frac{d\mathrm{PP}(t)}{dt} = k_1 \cdot \mathrm{DP} - (k_{-1} + k_f) \cdot \mathrm{PP}(t) \tag{8}$$

If we assume that the DP does not change size during the experiment, we get the general solution:

$$\mathrm{PP}(t) = \frac{k_1 \cdot \mathrm{DP}}{k_{-1} + k_f} - c \cdot e^{-(k_{-1}+k_f) \cdot t} \tag{9}$$

where *c* is an arbitrary constant.

If we insert the initial condition PP(0) = 0, we get the specific solution relevant for a pool recovery experiment:

$$\mathrm{PP}(t) = \frac{k_1 \cdot \mathrm{DP}}{k_{-1} + k_f} \left( 1 - e^{-(k_{-1}+k_f) \cdot t} \right) \tag{10}$$

where the PP size at equilibrium (t = ∞) is

$$\mathrm{PP}() = \frac{k_1 \cdot \mathrm{DP}}{k_{-1} + k_f} \tag{11}$$

### Model II – with a maximal primed pool size (release site model)

If we assume that the Primed Pool is limited by the number of release sites, we can modify (*Equation 8*) to include a maximal pool size, $PP_{max}$ (a constant). If we further assume that those release sites that are vacated by fusion, or depriming, are immediately available again for priming, the model becomes

$$\frac{d\mathrm{PP}(t)}{dt} = k_1 \cdot \mathrm{DP} \cdot (PP_{max} - \mathrm{PP}(t)) - (k_{-1} + k_f) \cdot \mathrm{PP}(t) \tag{12}$$

In reality, the release sites will take a while to recover after fusion (*Hosoi et al., 2009*); however, we shall consider the simpler situation. Note that $k_1$ has now changed unit from $s^{-1}$ to *release site*$^{-1}$ $s^{-1}$ (or $fF^{-1}s^{-1}$, in capacitance units).

For a pool recovery experiment (initial condition PP(0) = 0), we get the solution

$$\mathrm{PP}(t) = \frac{k_1 \cdot \mathrm{DP} \cdot PP_{max}}{k_{-1} + k_f + k_1 \cdot \mathrm{DP}} \left( 1 - e^{-(k_{-1}+k_f+k_1 \cdot \mathrm{DP}) \cdot t} \right) \tag{13}$$

where the time constant for recovery ($1/(k_{-1} + k_f + k_1 \cdot \mathrm{DP})$) now is sped up by the forward priming rate (this is a difference to *Model I*). The steady-state solution is

$$\mathrm{PP}(\infty) = \frac{k_1 \cdot \mathrm{DP} \cdot PP_{max}}{k_{-1} + k_f + k_1 \cdot \mathrm{DP}} \tag{14}$$

## Comparison of Models I and II (*Figure 4* and *Figure 4—figure supplement 1*)

We identified the depriming rate ($k_{-1}$) for WT and Syt-7 KO chromaffin cells by fitting a normalized version of (*Equation 10*) to a recovery curve, under the assumption that $k_f = 0$ (*Figure 4B–C*). When considering recovery at 60 ms after stimulation (corresponding approximately to the RRP), the results showed that $k_{-1}$ was increased by a factor of 2.12 in the Syt-7 KO (from 0.043 $s^{-1}$ in the WT to 0.091 $s^{-1}$ in the KO, *Figure 4B*). Comparison to the pool size obtained by the first stimulation

then made it possible to calculate the forward priming rate before stimulation, $k_1 \cdot \mathrm{DP}$ (from **Equation 11**), which was almost unchanged (3.21 fF/s in the WT and 3.07 fF/s in the Syt-7 KO). To take into account the slight overfilling during recovery in the WT, we assumed that this would result from a change in priming after stimulation. To calculate the $k_1 \cdot \mathrm{DP}$ after the stimulation, we multiplied $k_1 \cdot \mathrm{DP}$ by the fitted normalized plateau value (yielding 3.78 fF/s in WT and 1.90 fF/s in the Syt-7 KO). The recovery data at 600 ms were treated in the same way to identify $k_{-1}$, and $k_1 \cdot \mathrm{DP}$ before and after stimulation. All values are given in **Table 1**.

In order to understand how the presence of release sites would change the observation in the Syt-7 KO, we searched for a solution to Model II with unchanged recovery time constant and pool size in the WT case. For recovery assessed at 60 ms after stimulation, we first assumed that the RRP is at 90% capacity at rest ($k_f = 0$), which identifies $PP_{max}$ = 74.2 fF/0.9 = 82.4 fF. We now isolated $k_1 \cdot \mathrm{DP}$ and $k_{-1}$ from the two equations (where we again assume $k_f = 0$ during recovery):

$$k_{-1} + k_1 \cdot \mathrm{DP} = 0.043 \ \mathrm{s}^{-1} \tag{15}$$

$$\frac{k_1 \cdot \mathrm{DP} \cdot 82.4\mathrm{fF}}{k_{-1} + k_f + \ k_1 \cdot \mathrm{DP}} = 74.2 \ \mathrm{fF} \tag{16}$$

which yields $k_1 \cdot \mathrm{DP} = 0.039 \ \mathrm{s}^{-1}$ and $k_{-1}$ = 0.0043 $\mathrm{s}^{-1}$. Simulating a recovery experiment with these parameters in Model II yielded a curve identical to the one in Model I with WT parameters (**Figure 4E**). To understand the consequences of changing the depriming and priming rates, we increased $k_{-1}$ by a factor of 2.12 and fixed pre- and post-stimulation $k_1 \cdot \mathrm{DP}$ at the values found in the Syt-7 KO (**Table 1**). This yielded the calculated recovery curves for the Syt-7 KO in a cell with release sites (**Figure 4E**). Predicted recovery at 600 ms in a release site model was calculated in the same way (**Figure 4F**).

## Electron microscopy

Samples for the ultrastructural analysis of LDCV docking in chromaffin cells were prepared according to a published protocol (**Man et al., 2015**) with only minor modifications. Briefly, adrenal glands were removed from Syt-7 KO and WT littermate P0 pups and sectioned into 100µm-thick slices using a vibratome. Adrenal gland slices were allowed to recover for 15 min at 37°C in bicarbonate-buffered saline in the presence of 0.2 mM (+)-tubocurarine and then kept at room temperature in the same solution before cryofixation in external cryoprotectant (20% bovine serum albumin in bicarbonate-buffered saline) using a HPM100 HPF device (Leica). Freeze substitution was performed as previously published (**Rostaing et al., 2006**) and samples were embedded in EPON resin for 24 hr at 60°C. Ultrathin (60 nm, for 2D analysis) and semithin (350 nm, for 3D analysis) sections were collected onto Formvar-filmed, carbon-coated copper mesh grids. Ultrathin sections were poststained with uranyl acetate and lead citrate before imaging. Semithin sections were briefly incubated in Protein A conjugated to 15 nm gold particles (Cell Microscopy Center, Utrecht, The Netherlands). 2D- and 3D-EM imaging and the analysis of LDCV docking in chromaffin cells was performed exactly as previously published (**Man et al., 2015**).

## Calculation of total number and number of docked LDCVs in chromaffin cells

To calculate the total number of LDCVs per WT cell, we used the 2D analysis, together with the mean diameter of the vesicles, as determined from 3D-analysis (175.5 ± 4.1 nm). We use the cytoplasm density of vesicles ($\delta_v$v4.71 ± 0.18 vesicles/µm$^2$) and proceed to calculate the volume fraction of LDCVs in the cytoplasm. Each ultrathin section is 60 nm thick ($h$). We first assume that each vesicle profile within a slice occupy a cylinder-shape with a height of 60 nm and a diameter of 0.1755 µm (vesicular radius $r_{ves}$ = 0.08775 µm). The cylinder-shape occupies a volume of π x (0.08775 µm)$^2$ x 0.060 µm=0.00145 µm$^3$. The total volume of a 1 µm$^2$ area of the slice is 0.060 µm$^3$, and the (uncorrected) volume-fraction of LDCVs would be 4.71 × 0.00145 µm$^3$ / 0.060 µm$^3$ = 0.114. However, we need to correct for the spherical shape and tangential slicing of vesicles, which will lower the volume-fraction. The volume of a LDCV is (4/3) x π x (0.08775 µm)$^3$ = 0.002830 µm$^3$. The volume of the circumscribed cylinder is π x (0.08775 µm)$^2$ x (0.1755+0.060) µm = 0.00570 µm$^3$, where we have assumed that a LDCV is identified as such when 30 nm of a 60 nm section cuts the vesicle

tangentially, and thus we add $2 \times 30$ nm=60 nm to the cylinder height. Since cutting the vesicle at any point is equally likely, we apply the volume correction 0.00283 $\mu m^3$/0.00570 $\mu m^3$ = 0.497. Thus, the corrected volume-fraction of LDCVs is 0.114 × 0.497 = 0.0567. The radius of a chromaffin cell was determined from the mean cell capacitance measured in patch-clamp experiments (3.83 ± 0.057 pF). With a specific capacitance of $10^{-2}$ F/m$^2$, we get a plasma membrane (PM) area of 383 $\mu m^2$. Thus, the radius of a chromaffin cell is $r_{cell} = \sqrt{383\mu m^2/4\pi}$ = 5.52 $\mu$m. We assume that the nucleus' radius is half of the cell radius, i.e. 2.76 $\mu$m, which is comparable to the radius determined from the nucleus area in 2D EM micrographs (2.63 ± 0.06 $\mu$m), although the latter value will be affected by tangential slicing of the nucleus. The volume of the cytoplasm is then (4/3) π (5.52 $\mu$m)$^3$ - (4/3) π (2.76 $\mu$m)$^3$ = 704.5 $\mu m^3$ – 88.1 $\mu m^3$ = 616.4 $\mu m^3$. The total volume of LCDVs is therefore 0.0567 × 616.4 $\mu m^3$=34.95 $\mu m^3$, which yields 34.95 $\mu m^3$ / 0.002830 $\mu m^3$=12,334 LDCVs per Syt-7 WT cell. When assembling the considerations above into a single equation, the number of vesicles per cell ($N_v$) is given by

$$N_v = \frac{4\delta_v \cdot \pi \cdot r_{cell}^3 (1-f^3)}{3(2r_{ves}+h)} \tag{17}$$

where $\delta_v$ is the density of vesicles per $\mu m^2$ cytoplasm, $r_{cell}$ is the radius of the cell, $r_{ves}$ is the radius of a vesicle, $h$ is the thickness of an ultrathin section and $f$ is the ratio of nucleus to cell radius (here we used $f$ = 0.5). The corresponding numbers for Syt-7 KO are: LDCV diameter = 165.3 ± 3.4 nm, cytoplasm density of LDCVs = 4.70 ± 0.23 vesicles/$\mu m^2$, resting cell capacitance = 3.91 ± 0.062 fF, which yields a total of 13,270 LDCVs per Syt-7 KO cell.

The number of membrane-proximal LDCVs per cell ($n_a$, defined as vesicles within 40 nm of the cell membrane) can be calculated from the number of LDCVs per $\mu$m plasma membrane length ($n_l$) using the formula $n_a = n_l/(d_v + 0.06)$, where $d_v$ is the vesicle diameter (**Parsons et al., 1995**; **Plattner et al., 1997**). Our 3D-ET approach yields accurate estimates of $d_v$ (see above). For the Syt-7 WT, we estimated $n_l$ = 0.75 ± 0.05 vesicles/$\mu$m, which yields $n_a$ = 3.18 vesicles per $\mu m^2$. With a PM area of 383 $\mu m^2$ (see above), we get a total of ~1219 membrane-proximal (<40 nm) vesicles per cell. From 3D-ET, we know that 27 ± 2.5% of vesicles within 40 nm of the PM are docked (i.e. physically attached to the plasma membrane), which means Syt-7 WT cells have ~329 docked vesicles/cell. For the Syt-7 KO, we have estimated $n_l$ = 0.73 ± 0.05 vesicles/$\mu$m, which gives us $n_a$ = 3.24 vesicles per $\mu m^2$. With a PM area of 391 $\mu m^2$, we have a total of ~1267 vesicles within 40 nm of the PM. From 3D-ET 21 ± 2.5% are membrane-attached, which means that Syt-7 KO cells have ~266 docked vesicles/cell.

## Statistics

Sample sizes were not computed but based on the numbers that have typically been used in the literature before. The data are presented as mean ± SEM; N indicates the number of cells. Non-parametric Mann-Whitney or Kruskal-Wallis with post Dunn's test were applied for all capacitance measurements and release time constants. For other data, student's $t$-test or one-way ANOVA with post-hoc Tukey's test or Dunnett's test were applied on data with similar variances. The variances were analyzed with F test for two-sample data or with Bartlett's test for comparing more than two data samples. Heteroscedastic data were log transformed to satisfy the prerequisite of homogeneous variances. Non-parametric Mann-Whitney or Kruskal-Wallis with post Dunn's test were applied on data that failed to meet the criteria for parametric test after log-transformation.

## Acknowledgements

We thank Anne Marie Nordvig Petersen and Dorte Lauritsen for expert technical assistance. This investigation was supported by the University of Copenhagen 2016 (KU2016) excellence program, the Novo Nordic Foundation, the Danish Medical Research Council (all JBS), and the Lundbeck Foundation (JBS and PSP). We thank Erwin Neher for commenting on an earlier version of the manuscript.

## Additional information

### Competing interests

Nils Brose: Reviewing editor for eLife. The other authors declare that no competing interests exist.

### Funding

| Funder | Grant reference number | Author |
|--------|------------------------|--------|
| University of Copenhagen | KU2016 | Jakob Balslev Sørensen |
| Novo Nordisk Foundation | NNF19OC0058298 | Jakob Balslev Sørensen |
| Lundbeckfonden | R221-2016-1202 | Jakob Balslev Sørensen |
| Independent Research Fund Denmark | 0134-00141A | Jakob Balslev Sørensen |
| Lundbeckfonden | R34-A3740 | Paulo S Pinheiro Jakob Balslev Sørensen |

The funders had no role in study design, data collection and interpretation, or the decision to submit the work for publication.

### Author contributions

Bassam Tawfik, Formal analysis, Investigation, Writing - original draft, Writing - review and editing; Joana S Martins, Sébastien Houy, Cordelia Imig, Investigation, Writing - review and editing; Paulo S Pinheiro, Sonja M Wojcik, Investigation; Nils Brose, Supervision, Writing - review and editing; Benjamin H Cooper, Supervision, Investigation, Methodology; Jakob Balslev Sørensen, Conceptualization, Supervision, Funding acquisition, Writing - original draft, Project administration, Writing - review and editing

### Author ORCIDs

Bassam Tawfik (iD) http://orcid.org/0000-0003-1193-8494
Sébastien Houy (iD) http://orcid.org/0000-0003-3639-1931
Cordelia Imig (iD) http://orcid.org/0000-0001-7351-8706
Jakob Balslev Sørensen (iD) https://orcid.org/0000-0001-5465-3769

### Ethics

Animal experimentation: Mice were kept in an AAALAC-accredited stable at the University of Copenhagen operating a 12h/12h light/dark cycle with access to water and food ad libitum. Permission to keep and breed KO mice were obtained from the Danish Animal Experiments Inspectorate (permissions 2006/562-43 and 2018-15-0202-00157).

### Decision letter and Author response

Decision letter https://doi.org/10.7554/eLife.64527.sa1
Author response https://doi.org/10.7554/eLife.64527.sa2

## Additional files

### Supplementary files

• Transparent reporting form

### Data availability

All data generated or analysed during this study are or will be included in the manuscript and supporting files.

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
