## [Decision Letter]

**Acceptance summary:**

This comprehensive study examines the interaction of synaptotagmin-1 and synaptotagmin-7 in dense core vesicle fusion. The authors present extensive data demonstrating that synaptotagmin-7 and synaptotagmin-1 are sometimes not co-localized on the same vesicular structures. The authors also show that synaptotagmin-1 and -7 can act in a functionally independent manner.

**Decision letter after peer review:**

Thank you for submitting your article "Synaptotagmin-7 places dense-core vesicles at the cell membrane to promote Munc13-2- and Ca^2+^-dependent priming" for consideration by *eLife*. Your article has been reviewed by 3 peer reviewers, and the evaluation has been overseen by a Reviewing Editor and Gary Westbrook as the Senior Editor. The reviewers have opted to remain anonymous. The reviewers have discussed the reviews with one another and the Reviewing Editor has drafted this decision to help you prepare a revised submission.

Summary:

This is a comprehensive study that examines the interaction of synaptotagmin-1 and synaptotagmin-7 in dense core vesicle fusion. The authors present extensive data demonstrating that synaptotagmin-7 and synaptotagmin-1 are not co-localized on the same vesicular structures. The authors also show that both synaptotagmins can act in a functionally independent manner. However, these results – which would argue that these key molecules largely mark independent pools of vesicles that maintain RRP and SRP – show a striking contrast to other functional results that support interaction of the two calcium sensors.

Essential revisions:

1. The manuscript requires a rebalancing of its interpretations. Overall, non-overlapping localization of the two sensors is quite consistent with earlier results that demonstrated the divergence of synaptotagmin-7 and synaptotagmin-1 trafficking pathways as well as similar divergence of trafficking pathways that recover synchronously and asynchronously released vesicles (e.g. Virmani et al., 2003; Raingo et al., 2012; Li et al., 2017, Lin et al., 2020). Therefore, the authors are asked to re-organize their interpretation with an unbiased presentation of possible options. For instance, functional interactions between the two forms of release may simply be due factors extrinsic to the two vesicle pools. In addition, overexpression and loss-of-function manipulations interfere with these two sensors chronically, likely affecting vesicle biogenesis and vesicle identity processes in the long term. Therefore, they may not be simply interpreted as an acute effect on properties of fusion. Moreover, giving the well-documented roles of these two sensors on vesicle trafficking, it is highly likely that the apparent cross-talk between the two pools may occur during post-fusion steps.

2. Capacitance was analyzed according to RRP, SRP, taus, sustained rates, and total release. On lines 123-130 a connection is made between the taus and pools, but the fast tau in DKOs is presented with a caveat that there is hardly any RRP. If taus are a poor measure of pool sizes or release kinetics, what is the rationale for presenting them? Meanwhile, amperometry presumably represents a redundant measure of release that supports their capacitance measurements. But after Figure 1 amperometry is presented sporadically, and it is not clear what purpose this data serves.

3. The rationale for grouping data by the pre-stimulation calcium levels needs to be discussed in greater detail, and the way that this data was grouped should be addressed in the main text, or at least in the methods. When cytosolic calcium is ~200 nM, pharmacological inhibition of NSF-dependent de-priming increases the RRP in Syt-7 KO but not WT synapses. Similarly, phorbolesters increase the RRP and SRP in WT cells (but not KOs) when cytosolic calcium is ~200 nM. But when calcium is ~400 nM, phorbolesters increase the RRP, but not the SRP, in both genotypes. The authors should carefully assess their conclusion that Syt-7 performs certain functions in priming and depriming in 200 nM, but not 400 nM calcium.

4. The NEM experiments are confusing. The data in Figure 5A looks qualitatively different from other WT experiments. The amperometry data is noisy, shows little burst release after stimulation, does not mirror the kinetics of the capacitance traces, and the cumulative amperometry signal in WT/NEM appears to begin at ~.6 seconds. The NEM experiments with higher prestimulation calcium are referenced on Line 324 as being "below". This data is presented in Figure 5 – Supplement 1, and should be referenced explicitly. The conflicting results between Figure 5 and its supplement are barely discussed, leaving the reader confused as to what these experiments mean.

5. The authors use NEM to decrease depriming rates in Figure 5. In Figure 4 they assume that depriming rates are essential for the time course of RRP recovery. This argument would be strengthened by comparing the recovery with and without NEM, and would give credence to their argument that priming and depriming can be observed independently in chromaffin cells.

6. The descriptions of the antibodies used is confusing and conflicting. The description of Synaptic Systems antibodies in the methods does not match Figure 9 Supp 2 legend. In the methods section beginning at line 746, there are numerous typos, text duplications, and it is impossible to determine which antibodies were used for which experiments. This is important since the Σ-Aldrich antibody presumably targets the C2A domain, and the authors suggest that less staining correlates with lower expression levels for constructs that express C2A* mutants (Lines 232-4). Might this explain a lack of significance in the GFP signal seen for the various pHluorin-tagged Syt7 constructs in Figure 2 Supp 2 B? The use of this phluorin tag is only briefly mentioned in the methods.

7. The interpretation of the immunolabeling results is complicated by the fact that the tested anti Syt7 antibodies generate significant background staining in Syt7 ko cells (e.g. Figure 9b and Figure 9 Supplement 2b). In fact, the specificity of the Syt7 antibody cannot be judged form the western blot analysis (Figure 9 Supplement 2G) which depicts only a small fraction of the gel.

8. In the same line, the co-labelling experiment (confocal microscopy) against chromogranin and either Syt1 or Syt7 is not very convincing. The authors could use 3D-SIM (as used for Syt1 and Syt7 in Figure 9) to clarify to what extent either syt staining co-localizes with chromogranin.

9. Particularly worrisome is the observation that Syt7 and also Syt1 appear to accumulate in subcellular regions when Syt7 (or one of its truncated variants) is expressed in chromaffin cells (Figure 2 Supplement 2).

10. Several figures explore the limited colocalization of Syt-1 and Syt-7 on organelles, even though the key finding of Figure 9 has already been shown by Bendahmane et al., 2020. However, the discussion concludes that the 2 sensors "might co-localize on some or even most vesicles" (Line 594-5). If the 2 Syts operate cooperatively, one would expect significant colocalization. Perhaps it would clarify the situation if colocalization analysis was combined with CgA staining to focus on secretory granules? The lack of co-localization leaves room for alternative scenarios (i.e. Syt7 might promote the maturation of dense-core granules rather than their priming). Please comment and revise the text.

[Editors' note: further revisions were suggested prior to acceptance, as described below.]

Thank you for resubmitting your work entitled "Synaptotagmin-7 places dense-core vesicles at the cell membrane to promote Munc13-2- and Ca^2+^-dependent priming" for further consideration by *eLife*. Your revised article has been reviewed by 3 peer reviewers, and the evaluation has been overseen by a Reviewing Editor and Gary Westbrook as the Senior Editor. The manuscript has been improved but there is a remaining issue raised by reviewer 2 that needs to be addressed, as outlined below. Once we receive your revision, we will expeditiously proceed with acceptance.

Reviewer #1:

The authors have addressed my earlier comments diligently. I respect their position. I do not have concerns that would preclude publication.

Reviewer #2:

The authors have done a good job responding to most of the reviewers' concerns. They now give a more nuanced interpretation of their results, a more complete discussion of the rationale for their experiments, and improved description of methods. Some extraneous amperometry data was removed, simplifying the physiology figures and making the results easier to digest.

The only remaining concern is the use of IHC to determine the relative localization of the 2 Syt isoforms. The authors conclude that Syt1 and Syt7 do not overlap, but are instead located on the outside of the same DCVs. However, as presented, these results are hard to square with their interpretations. The new Figure 10 shows 3D-SIM images and colocalization analysis for Syt isoforms against CgA. Panel 10A/E each have 7 subpanels(!) with 3 scale bars which should be better referenced in the legend. 10A/E both focus on ROIs that depict Syt puncta dead center in a CgA spot. Yet the analysis in 10D/H and overall conclusion is that Syts are excluded from the center of CgA puncta. Presumably, this disparity is due to the 3D analysis they performed. But then why focus on the ROIs that contradict the point? Is there a way to use 3D depictions of the data, and more rigorous methods (random shuffling of puncta) to explore the association of Syts with the outside of CgA spots? SIM is prone to post-processing artifacts, and the axial resolution is still not as good as the lateral resolution, so I am not sure that 3D segmented-based localization can achieve their desired results. Overall, the authors have presented such varied data on the localization of Syts that I do not feel they can state decisively: "Syt-1 and Syt-7 are both closely associated with dense-core vesicles, and the majority of dense-core vesicles harbors both Syt-1 and Syt-7 spots."

Reviewer #3:

In the manuscript Tawfik et al. have investigated the role of synaptotagmin 7 in the secretory response of mouse chromaffin cells. With a combination of electrophysiological, fluorescence and electron microscopy techniques the authors demonstrate that the two Ca^2+^ sensing proteins Synaptotagmin 1 and Synaptotagmin 7 do not functionally overlap, but complement each other in facilitating Ca^2+^-dependent secretion. While Syt1 triggers the fast exocytosis of RRP vesicles, Syt7 appears to promote the slower exocytosis from the SRP and in addition enhances general vesicle priming, thereby increasing the overall secretion response.

The quality of the manuscript has significantly improved and the authors have answered all my questions. In particular, the newly added 3D SIM data illustrate the clear association of both, SytI and SytVII to the majority of chromogranin positive granules (Figure 10) resolving the discrepancy between functional interdependency and an apparently different subcellular localization (compare Figure 9).

Overall, the manuscript provides new insights into the function of Syt7 in neuroendocrine cells.

---

## [Author Response]

Summary:This is a comprehensive study that examines the interaction of synaptotagmin-1 and synaptotagmin-7 in dense core vesicle fusion. The authors present extensive data demonstrating that synaptotagmin-7 and synaptotagmin-1 are not co-localized on the same vesicular structures. The authors also show that both synaptotagmins can act in a functionally independent manner. However, these results – which would argue that these key molecules largely mark independent pools of vesicles that maintain RRP and SRP – show a striking contrast to other functional results that support interaction of the two calcium sensors.

We thank the reviewers for taking time to provide valuable suggestions that we have worked through carefully.

The major experimental change is that we have followed the suggestion in points 8 and 10 to perform 3D-SIM on stainings of Chromogranin A (CgA) vs Syt-1 and CgA vs Syt-7 using antibodies directed against the cytoplasmic domains of both synaptotagmins. This analysis showed striking results, as presented in Figure 10 of the revised manuscript. CgA-positive structures (dense core vesicles) are clearly distinct from both Syt-1 and Syt-7 structures highlighting the improved resolution of this imaging approach. The CgA-positive punctae corresponding to vesicles are larger in size, they are not exactly colocalized with Syt-1 or Syt-7 spots, and the number of Syt-spots is much higher than the number of CgA-vesicles. Using a 3D-distance estimation (center-to-center between CgA and Sytspots), we can show that both Syt-1 and Syt-7 positive punctae cluster just outside the CgA-vesicles (Figure 10D, 10H). The bell-shaped curve shows that Syt-positive punctae are excluded from the interior of the vesicle (as expected because the antibodies recognize the cytoplasmic part of the protein), but they cluster just outside the vesicle, roughly at the distance expected for a membrane-attached protein. We are not able to do triple labeling (CgA, Syt-1 and Syt-7) due to the lack of compatible antibodies, but the double-labeling shows that the large majority of vesicles are closely associated with at least one Syt-1 and one Syt-7 cluster. Thus, Syt-1 and Syt-7 cluster independently (as we still show in Figure 9, consistently with Bendahmane et al., 2020); but the clusters are found on or near the surface of CgA-vesicles. We discuss the interpretation in the Discussion. Note that the distribution is very similar for Syt-1 and Syt-7 (Figure 10D and 10H), indicating that they localize to vesicles in a similar manner.

Essential revisions:1. The manuscript requires a rebalancing of its interpretations. Overall, non-overlapping localization of the two sensors is quite consistent with earlier results that demonstrated the divergence of synaptotagmin-7 and synaptotagmin-1 trafficking pathways as well as similar divergence of trafficking pathways that recover synchronously and asynchronously released vesicles (e.g. Virmani et al., 2003; Raingo et al., 2012; Li et al., 2017, Lin et al., 2020). Therefore, the authors are asked to re-organize their interpretation with an unbiased presentation of possible options. For instance, functional interactions between the two forms of release may simply be due factors extrinsic to the two vesicle pools. In addition, overexpression and loss-of-function manipulations interfere with these two sensors chronically, likely affecting vesicle biogenesis and vesicle identity processes in the long term. Therefore, they may not be simply interpreted as an acute effect on properties of fusion. Moreover, giving the well-documented roles of these two sensors on vesicle trafficking, it is highly likely that the apparent cross-talk between the two pools may occur during post-fusion steps.

In the previous version, we had presented in the Discussion two possibilities for how Syt-1 and Syt-7 could cooperate: either directly (due to colocalization on vesicles), or indirectly. For the latter possibility, we had suggested that Syt-7 vesicles would fuse first and interact with Syt-1 vesicles before they fuse, which is an example of post-fusion interaction. We also, however, discussed wellknown data from chromaffin cells showing that SRP-vesicles mature to RRP vesicles before they fuse, which made the post-fusion interaction tenuous. Similar data have been reported from neurons (Taschenberger et al., 2016, PNAS 113:E4548-4578; Lee et al., 2012, PNAS 109:E765-774). With the new data, we now clarify that Syt-1 and Syt-7 are associated to CgA-containing vesicles in a similar manner, and that most CgA-vesicles are associated with both Syt-1 and Syt-7. We therefore reorganized the first part of the Discussion to reflect this fact. Regarding the four papers from the Kavalali-group, we cite the two papers that are concerned with Syt-7 (Virmani et al., 2003; Li et al., 2017). We do not cite the two papers concerned with VAMP4, although we understand the argument that they support the idea of fusion machineries localized to different vesicles. Given our new data and the already well-known different localization of Syt-7 in chromaffin cells (granules) and neurons (plasma membrane), we would like to concentrate this point of our discussion on chromaffin cells, to not confuse the systems. Dense-core vesicles and synaptic vesicles are formed via different biosynthetic pathways, and the situation may be different. We actually have preliminary data showing VAMP4 accumulating at the TGN in chromaffin cells, although part of the protein might be on vesicles, but going into this is beyond the scope of our paper. Regarding the argument that overexpression and knockout interferes with vesicle biogenesis and identity, we would like to point out that no major conclusion in the manuscript depends solely on overexpression; the conclusions can be largely made by comparing Syt-7 KO and WT cells, although we used overexpression to demonstrate rescue (see also answer to point 9 below). Moreover, note that the effect of Syt-7 on calcium-dependent priming is complete 20 s after raising prestimulation calcium concentrations (and depends on intact calcium binding sites), the effect of Syt-7 on repriming after emptying requires a similar timespan, and the effect of phorbolester on priming in the presence (but not the absence) of Syt-7 requires only a minute of treatment with phorbolester. Thus, our data show that the effect of Syt-7 on priming is acute and therefore not likely to be due to indirect biosynthetic changes. Finally, note that our high-resolution EM data shows the same number of vesicles in Syt-7 KO and WT cells, and no noticeable changes in morphology. We have inserted a comment in the Discussion about the acute effects of Syt-7.

2. Capacitance was analyzed according to RRP, SRP, taus, sustained rates, and total release. On lines 123-130 a connection is made between the taus and pools, but the fast tau in DKOs is presented with a caveat that there is hardly any RRP. If taus are a poor measure of pool sizes or release kinetics, what is the rationale for presenting them? Meanwhile, amperometry presumably represents a redundant measure of release that supports their capacitance measurements. But after Figure 1 amperometry is presented sporadically, and it is not clear what purpose this data serves.

Taus have been used for many years to measure release kinetics, whereas amplitudes are used to measure pool sizes, based on the assumption of single-exponential emptying of a vesicle pool. The reason we place a caveat for the interpretation of the RRP in the Syt-1/7 Double KO is that the amplitude is so low that it corresponds to only a few (3-4) vesicles, which is not a lot to base an estimate on. Any small imperfection in the capacitance measurements, and the background noise on the measurement, could affect such a number, and the associated tau is thus not that well determined. This does not imply that taus are generally a poor measure. In all other cases, amplitudes are larger and taus are therefore better determined. We have now explained in more detail what the reason is for placing the caveat.

Regarding amperometry, we performed these measurements in parallel, because it has been claimed by some (Zhang et al., 2019, but see Segovia et al., 2010) that kiss-and-run fusion events are upregulated in the Syt-7 KO, which could cause a reduction of overall capacitance in the Syt-7 KO, without a similar reduction in catecholamine release (because some vesicles would release catecholamines, but then close the fusion pore). We explained this in the original manuscript, and we have tried to clarify further: “Kiss-and-run fusion would cause catecholamine release without net capacitance change. We therefore performed amperometric measurements in parallel with capacitance measurements; these measurements cannot distinguish between RRP and SRP fusion, due to the diffusional delay before catecholamines are recorded at the amperometric fibre, but importantly the total measured amperometric charge and capacitance changed in parallel (Figure 2A,F bottom panels; quantification of total amperometric release in Figure 2 —figure supplement 1), indicating that vesicle fusion and adrenaline release were similarly affected by the presence of Syt-7.” Also, conditions are known where catecholamine release is inhibited, while capacitance increases is unchanged, because fusing vesicles are not releasing catecholamines. Due to the ability to distinguish between the RRP and the SRP together with the fact that amperometry only detects a (variable) fraction of the catecholamines released from a cell capacitance is the preferred measurement, but we quantified the total amperometric release and placed them in supplementary figures. As it turns out, in all cases that we examined in this manuscript, amperometry changed in parallel with capacitance, so it might seem that these data were not used, but in fact they are important. We have now placed some more comments on the amperometry in the Result section, making these points more explicitly.

3. The rationale for grouping data by the pre-stimulation calcium levels needs to be discussed in greater detail, and the way that this data was grouped should be addressed in the main text, or at least in the methods. When cytosolic calcium is ~200 nM, pharmacological inhibition of NSF-dependent de-priming increases the RRP in Syt-7 KO but not WT synapses. Similarly, phorbolesters increase the RRP and SRP in WT cells (but not KOs) when cytosolic calcium is ~200 nM. But when calcium is ~400 nM, phorbolesters increase the RRP, but not the SRP, in both genotypes. The authors should carefully assess their conclusion that Syt-7 performs certain functions in priming and depriming in 200 nM, but not 400 nM calcium.

We agree that we did not explain this properly, and we have now inserted an explanation of the pre-stimulation calcium levels used. The point of the ‘high prestimulation calcium” condition is to adjust the prestimulation calcium to around the concentration, where priming is maximal. Since the plot of the primed vesicles pools versus prestimulation calcium typically is a bell-shaped curve (Voets, 2000; at higher calcium-concentration the primed vesicle pools are depleted due to ongoing fusion), this concentration should be below the secretion threshold, and in untreated wildtype cells, this corresponds to 600-800 nM calcium. However, we found that in phorbolester treated cells, the secretion threshold is lower, and secretion becomes massive already at 500 nM. This is consistent with measurements of release rates in the calyx of Held in the presence of phorbolesters (Lou et al., 2005, Nature 435: 497-501). As a result, the point of maximal priming is moved to lower calcium concentrations. Therefore, for these experiments the ‘high prestimulation calcium’ condition was set at 400 nM, since it would make no sense to measure on a cell where the vesicle pools had been depleted before measurements. We have now explained this in the text.

Thus, it is correct that we find a difference between 200 nM and 400 nM calcium for phorbolester treatment. However, the difference is not so large as alluded to by the reviewers: the RRP is significantly increased in the Syt-7 KO at 200 nM prestimulation calcium, and the increase in the SRP is almost significant (p=0.0506), as we note in the text. At higher prestimulation calcium (400 nM), the SRP is not affected by phorbolester in the WT or Syt-7 KO, probably because another calciumdependent process can take over from Syt-7. We think we are quite careful in the interpretation of these data, as we conclude that at low prestimulation calcium “Syt-7 assists in ubMunc132/phorbolester-dependent priming” and “the combination of Syt-7 KO with high prestimulation [Ca^2+^] and other manipulations to increase Munc13-priming (ubMunc13-2, PMA) allow priming in the Syt-7 KO, which is consistent with the presence of several priming pathways in chromaffin cells (Liu et al., 2010). If the main function of Syt-7 is to deliver vesicles to the membrane (and to Munc13) other calcium sensors that carry out similar functions might compensate for the absence of Syt-7”. Thus, we do not conclude that syt-7 does not perform functions in priming at higher prestimulation [Ca^2+^], but that other factors can partly compensate.

4. The NEM experiments are confusing. The data in Figure 5A looks qualitatively different from other WT experiments. The amperometry data is noisy, shows little burst release after stimulation, does not mirror the kinetics of the capacitance traces, and the cumulative amperometry signal in WT/NEM appears to begin at ~.6 seconds. The NEM experiments with higher prestimulation calcium are referenced on Line 324 as being "below". This data is presented in Figure 5 – Supplement 1, and should be referenced explicitly. The conflicting results between Figure 5 and its supplement are barely discussed, leaving the reader confused as to what these experiments mean.

We thank the reviewer for pointing out that the amperometric data presented in Figure 5 looked strange. We went through these data again and indeed realized that we had made a mistake. The amperometric current in an uncaging experiment is ‘contaminated’ by the fact that light (from the flash lamp, or the monochromator used to measure calcium concentrations) elicits an amperometric current in some cases (it depends on the exact shape and placement of the amperometric fibre). Therefore, we have to correct these data after measurement by applying an identical light stimulus to the fibre in the absence of a cell, and subtract them from measurements (explained in Materials and methods). However, that had not been done in all cases here, and consequently there was a strange repetitive ‘noise’ like appearance on the amperometric current, which is due to illumination from the monochromator. In addition, it appears that an insensitive fibre was used for part of these measurements (fibres can lose sensitivity abruptly during measurements). We have therefore removed the amperometric currents from this particular set of measurements.

Regarding the NEM data with higher prestimulation calcium, the ‘below’ is in reference to a paragraph, where higher prestimulation calcium recordings are presented for NEM, ubMunc13-2 overexpression and phorbolester data. In that section, we describe and interpret the NEM-data from higher prestimulation calcium and references Figure 5 – supplement 1 directly, including an interpretation of the findings.

5. The authors use NEM to decrease depriming rates in Figure 5. In Figure 4 they assume that depriming rates are essential for the time course of RRP recovery. This argument would be strengthened by comparing the recovery with and without NEM, and would give credence to their argument that priming and depriming can be observed independently in chromaffin cells.

This would be a nice experiment, but the NEM-experiments are quite tricky, since NEM has to be present for long enough that NSF is blocked (and the presumed effect of NSF on depriming is therefore absent, leading to an increase in secretion), but not for so long that cis-SNARE-complexes accumulate, which will then reduce secretion (because the cell runs out of options to create transcomplexes). We found that incubating the cells for 60-100 s (NEM is in the pipette) has a rescuing effect on Syt-7 KO at low prestimulation calcium. However, this would be very hard to combine with a recovery experiment, because that experiment requires us to wait for different lengths of time

(between 4 and 115s, Figure 4—figure supplement 4). Thus, any effect or non-effect of NEM would be very hard to interpret. From a more practical point of view, the recovery experiment is a huge amount of work. We measured from 92 Syt-7 KO cells and 97 WT cells (Figure 4—figure supplement 4) to generate the data in Figure 4. Repeating this would take months (we can get 1 or 2 knockout pups per week, and measure from maybe 10-15 KO cells in a good week, and a similar number of WT cells). We have therefore not followed up on this suggestion, given the large amount of work and the unclear interpretation.

6. The descriptions of the antibodies used is confusing and conflicting. The description of Synaptic Systems antibodies in the methods does not match Figure 9 Supp 2 legend. In the methods section beginning at line 746, there are numerous typos, text duplications, and it is impossible to determine which antibodies were used for which experiments. This is important since the Σ-Aldrich antibody presumably targets the C2A domain, and the authors suggest that less staining correlates with lower expression levels for constructs that express C2A* mutants (Lines 232-4). Might this explain a lack of significance in the GFP signal seen for the various pHluorin-tagged Syt7 constructs in Figure 2 Supp 2 B? The use of this phluorin tag is only briefly mentioned in the methods.

Thank you for pointing out our mistake in Figure 9 Supp 2 – as well as a similar mistake in the Results section. We have now corrected the information. In Figure 2 Supp 2 we used the two Synaptic Systems antibodies (rabbit polyclonal α-Syt7, Synaptic Systems 105173, and mouse monoclonal α-Syt1 Synaptic Systems 105011) to assess overexpression. Thus, we did not use the Σ-Aldrich antibody here, and the SySy antibody is directed toward the linker of Syt-7 (between transmembrane domain and C2A). Please note that the difference between GFP signals is entirely consistent with the difference between the Syt-7 signals. We did not use the pHluorin tag for other purposes than verification of expression, which we write in the Materials and methods. Regarding the Materials and methods section, we had written out a list of the different conditions, which was rather hard to read. We have now rewritten each condition to improve clarity.

7. The interpretation of the immunolabeling results is complicated by the fact that the tested anti Syt7 antibodies generate significant background staining in Syt7 ko cells (e.g. Figure 9b and Figure 9 Supplement 2b). In fact, the specificity of the Syt7 antibody cannot be judged form the western blot analysis (Figure 9 Supplement 2G) which depicts only a small fraction of the gel.

This Synaptic Systems antibody unfortunately has some background in the syt-7 KO cells. We therefore also performed staining with another antibody (Σ-Aldrich MABN655, which was used in a recent paper on Syt-7 (Barthet et al., 2018, Nature Commun. 9:4780)). This antibody has less background and yielded similar results (Figure 9– supplement 2D-F). The new stainings we performed in revision (Figure 10) are with the Σ-Aldrich antibody. We do not know why the SySy antibody has background, and we cannot rule out limited specificity of the antibody, so we have removed the remark. Please note that the residual fluorescence is rather homogenous, and so if we average around Syt-1 positive peaks in the Syt-7 KO, the residual fluorescence in the Syt-7 channel displays no colocalization with Syt-1 (Figure 9 – supplement 3 in the first manuscript version; this figure is removed in the revised version), so we do not make a mistake based on this. Let us finally remark that in our manuscript we have quantified the residual fluorescence in the Syt-7 KO, which we think is important (and honest). We note that not all manuscripts contain this quantification, making it hard to compare to previous work. We also note that our staining has other, internal controls, such as a lack of punctate staining in the nucleus (in some cases lack of specificity causes punctate-like staining in the nucleus). Overall, we think we cannot do much better, and our conclusions are unlikely to be affected by this problem.

8. In the same line, the co-labelling experiment (confocal microscopy) against chromogranin and either Syt1 or Syt7 is not very convincing. The authors could use 3D-SIM (as used for Syt1 and Syt7 in Figure 9) to clarify to what extent either syt staining co-localizes with chromogranin.

Thank you for pushing us here! We have repeated these stainings and subjected them to 3DSIM, and present the data in the new Figure 10. Please see our answer above, below ‘Summary’.

9. Particularly worrisome is the observation that Syt7 and also Syt1 appear to accumulate in subcellular regions when Syt7 (or one of its truncated variants) is expressed in chromaffin cells (Figure 2 Supplement 2).

The accumulation is most striking when C2A or C2B mutations are expressed, but it is also seen when expressing the WT protein. We think it is caused by the approximately two-fold overexpression. We note this in the main text and we think it is important to report these data, as several labs are using similar or identical constructs. As far as the conclusions of our manuscript are concerned, please note that our Syt-7 WT construct can support stand-alone fusion (Figure 1: expresssion in double KO), and can rescue kinetics and amplitude in the Syt-7 KO (Figure 2), and overrescue priming (Figure 3). Moreover, please note that all conclusions that we make in the manuscript are supported by direct comparison between the Syt-7 WT and Syt-7 KO condition; overexpression supports some of those conclusions, but is not necessary to establish them. Indeed, we only used overexpression to bolster Syt-7’s involvement in calcium-dependent priming and fusion kinetics (Figure 2-3), which is already clear from comparing Syt-7 KO and WT. Syt-7 overexpression was not used in establishing Syt-7 effects on forward and reverse priming rates (Figure 4: no overexpression), the effect of N-ethylmaleimide (Figure 5: no overexpression), phorbol ester (Figure 6: no overexpression), ubMunc132 (Figure 7: no overexpression of Syt-7), and 3D electron tomography (Figure 8: no overexpression). The analysis of Syt-1 and Syt7 overlap with each other and CgA (Figure 9 and Figure 10) also did not include overexpression data. We conclude, therefore, that these accumulations do not affect our conclusions.

10. Several figures explore the limited colocalization of Syt-1 and Syt-7 on organelles, even though the key finding of Figure 9 has already been shown by Bendahmane et al., 2020. However, the discussion concludes that the 2 sensors "might co-localize on some or even most vesicles" (Line 594-5). If the 2 Syts operate cooperatively, one would expect significant colocalization. Perhaps it would clarify the situation if colocalization analysis was combined with CgA staining to focus on secretory granules? The lack of co-localization leaves room for alternative scenarios (i.e. Syt7 might promote the maturation of dense-core granules rather than their priming). Please comment and revise the text.

Indeed, the reviewer is right. We have performed these stainings in 3D-SIM and present them in Figure 10. Please see our answer above, below ‘Summary’.

[Editors' note: further revisions were suggested prior to acceptance, as described below.]

Reviewer #2:The authors have done a good job responding to most of the reviewers' concerns. They now give a more nuanced interpretation of their results, a more complete discussion of the rationale for their experiments, and improved description of methods. Some extraneous amperometry data was removed, simplifying the physiology figures and making the results easier to digest.The only remaining concern is the use of IHC to determine the relative localization of the 2 Syt isoforms. The authors conclude that Syt1 and Syt7 do not overlap, but are instead located on the outside of the same DCVs. However, as presented, these results are hard to square with their interpretations. The new Figure 10 shows 3D-SIM images and colocalization analysis for Syt isoforms against CgA. Panel 10A/E each have 7 subpanels(!) with 3 scale bars which should be better referenced in the legend. 10A/E both focus on ROIs that depict Syt puncta dead center in a CgA spot. Yet the analysis in 10D/H and overall conclusion is that Syts are excluded from the center of CgA puncta. Presumably, this disparity is due to the 3D analysis they performed. But then why focus on the ROIs that contradict the point? Is there a way to use 3D depictions of the data, and more rigorous methods (random shuffling of puncta) to explore the association of Syts with the outside of CgA spots? SIM is prone to post-processing artifacts, and the axial resolution is still not as good as the lateral resolution, so I am not sure that 3D segmented-based localization can achieve their desired results. Overall, the authors have presented such varied data on the localization of Syts that I do not feel they can state decisively: "Syt-1 and Syt-7 are both closely associated with dense-core vesicles, and the majority of dense-core vesicles harbors both Syt-1 and Syt-7 spots."

We appreciate the concern. We have chosen new example vesicles for ROI2, which better represent the conclusion of the distance measurement. The scale bars are clearly set out in the Figure legend. We have also included a distance-estimation performed on the same data after inversion of the CgA-stack top-to-bottom (orange bar diagrams in Figure 10D and H). This partly randomizes positions (although not for vesicles placed in dead-center of the stack) and therefore provides an estimate for what the distances would be without any specific association between Syt and CgA. We preferred this method, because it preserves the position of the nucleus in the cell, whereas flipping left-to-right would often shift the nucleus from one side to the other, resulting in an overestimate of the distances after randomization. We have finally toned down our conclusions and replaced the sentence referenced above with: “Thus, Syt-1 and Syt-7 displays poor colocalization as they largely localize to different clusters, but they are both associated with dense-core vesicles, consistent with the localization of both Syt-1 and Syt-7 to a majority of dense-core vesicles. This does not rule out that some vesicles could harbor only or predominantly one of the two Syt isoforms.”